# VEGFR2 is required for VEGF-C–VEGFR3–PI3Kα-mediated sprouting lymphangiogenesis

Hans Schoofs [1,6], Yan Zhang[1,2,3], Henrik Ortsäter[1], Mariya Lytvyn[4], Rui Benedito [4,5] & Taija Mäkinen [1,2,3] ✉

Lymphatic vessels are essential for tissue homoeostasis and their growth is regulated by vascular endothelial growth factor C (VEGF-C) signalling through VEGFR3. However, how VEGF-C balances lymphatic endothelial cells (LECs) proliferation versus sprouting to ensure functional vessel formation has remained unclear. Using high-fidelity conditional genetics and receptor-specific ligands, we uncover a requirement for the alternative receptor VEGFR2 in VEGF-C-VEGFR3–driven lymphatic vessel sprouting. While activation of VEGFR2 alone fails to induce lymphangiogenesis, VEGFR2 loss abolishes LEC sprouting, but not proliferation, in response to VEGF-C. In contrast, deletion of the VEGFR3 downstream effector PI3Kα completely abrogates lymphangiogenesis. VEGFR2 is activated and found in proximity to VEGFR3 in LECs in vivo, with PI3Kα controlling their relative cell-surface availability and VEGF-C increasing VEGFR2 relative to VEGFR3, thereby priming LECs for sprouting. This receptor coordination balances VEGF-C-driven proliferative and sprouting responses, coupling LEC expansion to vessel growth, ensuring the formation of functional lymphatic networks.

The lymphatic vasculature is composed of a network of blind ended vessels, which take up fluid from the interstitial space and return it to the blood circulatory system. Emerging evidence indicates important functions of lymphatic vessels in the regulation of tissue homoeostasis and regeneration[1,2]. Accordingly, deregulated growth of lymphatic vessels is associated with several pathological conditions, such as chronic inflammation and cancer, thereby providing a potential target for therapeutic intervention.

The key regulator of both physiological and pathological lymphangiogenesis is vascular endothelial growth factor C (VEGF-C), which binds to its receptor VEGFR3 and to the co-receptor neuropilin 2 (NRP2) on lymphatic endothelial cells (LECs)[3]. After proteolytic processing, VEGF-C gains the ability to additionally bind and activate

VEGFR2, the major angiogenic receptor in blood endothelial cells (BECs), which is also abundantly expressed in LECs of both developing and mature vasculature[4]. Despite the essential function of VEGFR2 for early embryonic blood vascular development[5], conditional deletion of endothelial Vegfr2 during postnatal development has revealed relatively modest effects on blood vessel morphogenesis[6–10]. Recent studies have indicated that this is likely due to inefficient genetic deletion of the Vegfr2[flox] allele[8], and a disadvantage for Vegfr2-deleted cells compared to those maintaining VEGFR2 expression, with the latter outcompeting the mutant ECs during angiogenic growth[11]. Currently available data from LEC-specific gene deletion experiments largely suggest that VEGFR2 does not have a significant role during lymphatic vessel development or growth[9,10]. Studies using different VEGF family

[1]Department of Immunology, Genetics and Pathology, Uppsala University, Uppsala, Sweden. [2]Wihuri Research Institute, Helsinki, Finland. [3]Faculty of Medicine, University of Helsinki, Helsinki, Finland. [4]Molecular Genetics of Angiogenesis Group, Centro Nacional de Investigaciones Cardiovasculares (CNIC), Melchor Fernández Almagro, Madrid, Spain. [5]Max Planck Institute for Molecular Biomedicine, Department of Functional Genetics, Münster, Germany. [6]Present address: Division of Molecular Pathology, Oncode Institute, The Netherlands Cancer Institute, Amsterdam, the Netherlands. ✉e-mail: taija.makinen@helsinki.fi

ligands and their receptor-specific forms have further indicated that although VEGFR2 does not drive robust sprouting, it can selectively promote lymphatic vessel enlargement in postnatal vasculature in vivo[4,6,12]. In contrast to these in vivo findings suggesting that VEGFR2 is largely dispensable for lymphangiogenesis, in vitro studies have shown that VEGFR2 potently promotes LEC migration, proliferation and survival through the downstream PI3K/AKT and ERK1/2 pathways when stimulated by high doses of VEGF and/or VEGF-C[6]. However, it remains unclear whether VEGFR2 can be sufficiently activated in vivo in lymphatic endothelium by physiological levels of these ligands, and whether this activation induces downstream signalling beyond the signals already provided by VEGFR3. In cultured LECs, VEGF-C additionally induces heterodimerisation of VEGFR2 and VEGFR3, a configuration with a distinct pattern of phosphorylation as compared to a VEGFR3 homodimer activation[13]. Interestingly, VEGF-C/VEGFR3-mediated activation of AKT was found to depend on the presence of VEGFR2[14], further suggesting distinct functional outputs between VEGFR3 homodimers and VEGFR2/3 heterodimers in vitro. Given the ability of VEGFR2 to promote LEC signalling and function in vitro, it is intriguing that no major role for VEGFR2 has been reported in LECs in vivo. In this context, it is also noteworthy that most in vitro studies have utilized recombinant mature proteolytically cleaved VEGF-C, which, unlike the native uncleaved full-length VEGF-C protein, is capable of binding VEGFR2. However, the bioavailability and stability of the fully processed form in vivo is unknown.

Here, we investigate the role of VEGFR2 in lymphatic development in vivo using high-fidelity conditional genetics in combination with receptor-specific VEGF family ligands. We uncover a previously unknown critical function of VEGFR2 in VEGF-C-induced lymphatic vessel sprouting, both during development and in neo-lymphangiogenic growth in adult tissues. Our study suggests that for optimal therapeutic modulation of lymphangiogenesis, either its promotion or inhibition, both VEGFR2 and VEGFR3 pathways should be considered. It also raises an interesting possibility of repurposing anti-angiogenic therapies targeting VEGFR2 as anti-lymphangiogenic drugs in conditions of pathological excessive lymphangiogenesis.

## Results

### VEGFR2 activation does not promote lymphangiogenesis

To elucidate the role of VEGFR2 in lymphatic endothelium, we first re-examined the cellular effects reported for the activation of VEGFR2 and VEGFR3 in LECs in vivo, utilizing receptor-selective ligands. In line with previous findings[15], intradermal injection into the ear pinna of juvenile mice of adeno-associated viral vectors (AAVs) encoding native full-length VEGF-C, which becomes proteolytically processed in vivo and can bind VEGFR2, VEGFR3 and their heterodimer (Fig. 1a), induced extensive lymphatic hyperplasia. This was characterized by a chaotic and highly branched network of LYVE1+ lymphatic capillaries two weeks after AAV injection (Fig. 1b). Similarly, collecting vessels showed sprouting and ectopic expression of LYVE1 (Fig. 1c), making it difficult to distinguish these vessels. AAV-mediated overexpression of the VEGFR3-specific ligand VEGF-C C156S resulted in an apparent increase in the diameter of lymphatic capillaries without affecting their branching (Fig. 1b) or the morphology of collecting vessels (Fig. 1c,d). In contrast, specific activation of VEGFR2 by VEGF-A164 overexpression increased the diameter of lymphatic capillaries and collectors, but did not induce sprouting or expansion of lymphatic capillaries (Fig. 1b-d), consistent with previous studies, which reported only minor sprouting after adenoviral VEGF-A164 expression[4,12]. As expected, VEGF-A164 also caused pronounced blood vessel hyperplasia, while neither form of VEGF-C had an observable effect on the blood vasculature during the two-week observation period (Fig. 1b). These results, consistent with previous findings[4,12], suggest that selective VEGFR2 activation promotes lymphatic vessel enlargement but not sprouting. However, given the well-established role of VEGF/VEGFR2 in promoting blood

vessel permeability, we hypothesized that the observed lymphatic vessel phenotype may be secondary to leakage-associated inflammation and subsequent recruitment of VEGF-C producing macrophages[16,17].

To circumvent VEGF-induced vascular leakage and hyperplasia, we selectively deleted *Vegfr2* in blood vascular endothelial cells using the BEC-specific *Vegfr1-CreER^T2* line[18] in combination with a faithful Cre-reporter *R26-iSuRe-HadCre*[11] two weeks before AAV-VEGF-A164 injection (Fig. 1e, Supplementary Fig. 1a). Potential secondary effects caused by VEGF-induced recruitment of VEGF-C-producing macrophages[17] were additionally blocked by systemic administration of AAVs encoding the soluble VEGF-C-trap[19] one week prior to AAV-VEGF-A164 injection (Fig. 1e). Successful overexpression of VEGF-A164 was verified by ELISA analysis of tissue lysates (see methods). This approach thus allows direct interrogation of the effect of VEGFR2 homodimer activation in lymphatic endothelium. Whole-mount immunofluorescence confirmed VEGFR2 protein depletion specifically in BECs in homozygous *Vegfr2^flox/flox* mutant mice without apparent vascular defects (Supplementary Fig. 1b). Analysis of the ears from mutant mice two weeks after AAV-VEGF-A164 revealed an abrogated angiogenic response, with only small focal vascular lesions, likely arising from residual BECs that escaped recombination and remained responsive to VEGF-A164 (Fig. 1e). Interestingly, lymphatic vessel morphology (Fig. 1e) and diameter (Fig. 1d) remained unchanged in mutant compared to control skin, suggesting lymphatic vessel enlargement induced by VEGF overexpression in wild type mice are secondary to the observed blood vascular hyperplasia, leakage and macrophage recruitment.

Taken together, these results show that, opposite to its effect described in vitro[14], sole activation of lymphatic endothelial VEGFR2 by VEGF-A164 does not elicit a lymphangiogenic response in adult dermal vasculature in vivo. Yet, the distinct LEC responses observed upon overexpression of the native VEGF-C, which activates both VEGFR2 and VEGFR3 receptors and their heterodimers, compared to the VEGFR3 homodimer-specific form, VEGF-C C156S, suggest that VEGFR2 plays a role in modulating the VEGF-C response.

### *Vegfr2* deletion is incompatible with lymphatic sprouting

Next, we re-assessed the role of VEGFR2 during developmental lymphangiogenesis by analyzing the dermal lymphatic vasculature of the ear pinna. Superficial dermal lymphatic capillaries of the dorsal ear skin form postnatally through VEGF-C-dependent growth[20,21] from a primary vascular plexus present at postnatal day (P) 1 (Fig. 2a). This plexus undergoes lymphangiogenic sprouting towards the superficial layer of the dermis at P3 (Fig. 2a), giving rise to a mature two-layered network of lymphatic capillaries and collecting vessels by P21[22,23]. Since even small amounts of VEGFR2 are able to sustain developmental angiogenesis[8], we maximized gene targeting efficiency using mice carrying one conditional *Vegfr2^flox* allele in combination with a constitutive germline null allele (Fig. 2b). The mice were further crossed with LEC-specific *Prox1-CreER^T2* mice[24], and the *R26-mTmG* reporter[25] to track Cre-targeted cells. Gene deletion was induced by administering tamoxifen during the early postnatal period of active LEC proliferation and vascular growth (daily between P2-P5/7) (Fig. 2c) or extended until analysis (every second day between P2-P20) (Fig. 2d).

Whole-mount Immunofluorescence of the ear skin showed apparently normal development of the lymphatic vasculature in both mutant and littermate control mice using either the tamoxifen administration regime (Fig. 2c,d). GFP expression confirmed efficient and specific targeting of the entire lymphatic vasculature. However, while collecting vessels in the mutant mice showed efficient depletion of the VEGFR2 protein, lymphatic capillaries and in particular, the capillary tips were often composed of cells that still retained VEGFR2 expression (Fig. 2c,e). Since the *R26-mTmG* and *Vegfr2^flox* alleles are located at different loci, their recombination occurs independently. The observed selection and selective expansion of GFP+ LECs that

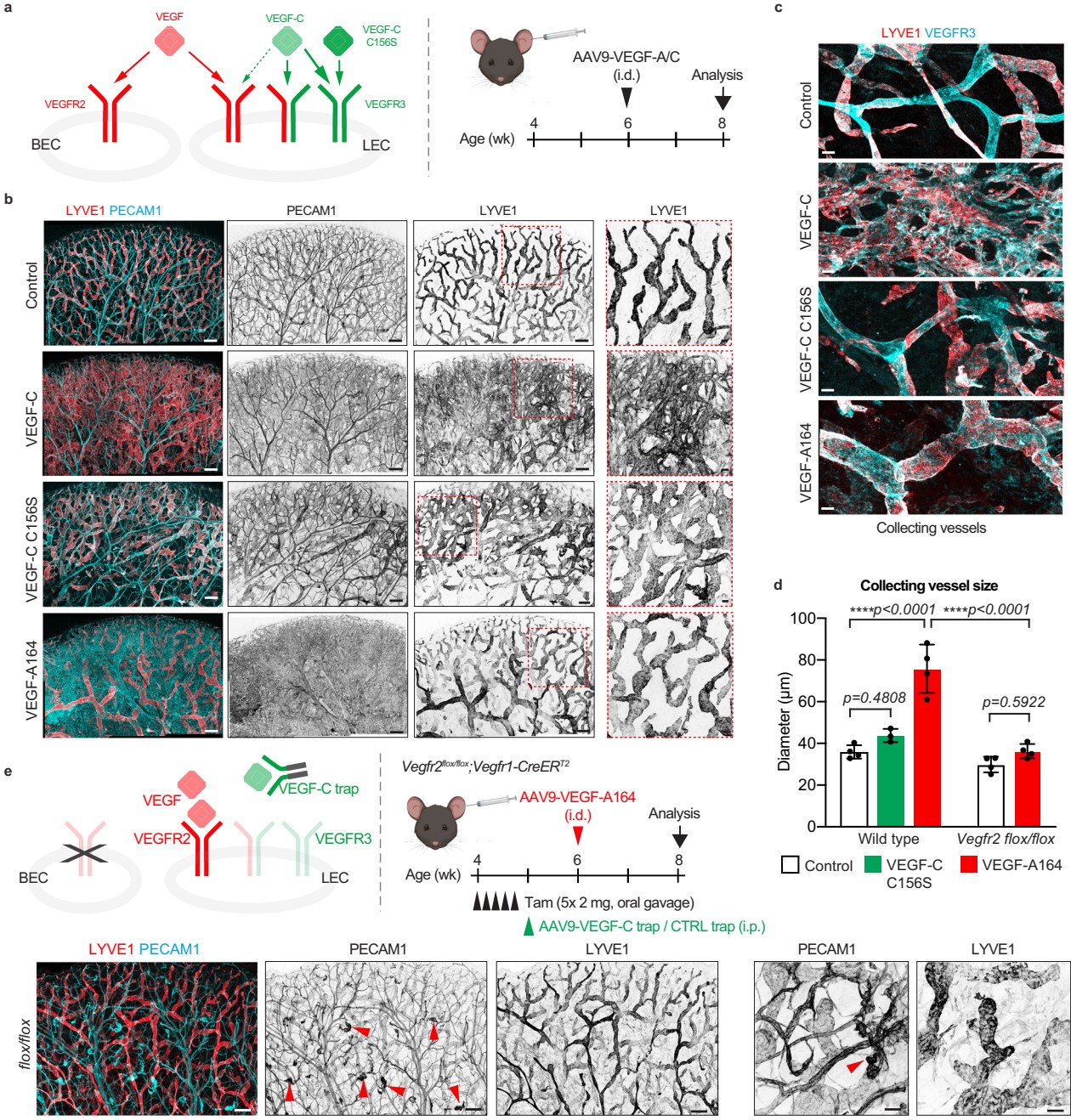

**Fig. 1 | Effects of selective VEGFR2 and/or VEGFR3 activation on postnatal lymphangiogenesis. a** VEGF-VEGFR interactions and experimental scheme for AAV-mediated overexpression of VEGF ligands in the ear. Whole mount immuno-fluorescence of mouse ear dermis two weeks after injection of AAVs encoding different VEGFs showing their different effects on lymphatic capillaries (**b**) and collecting vessels (**c**). Dashed boxed areas are magnified on the right (**b**). **d** Diameter of collecting vessels in AAV-treated mice, represented as mean ± s.d. (WT: $n = 4$ (Control), 3 (VEGF-C C156S), 4 (VEGF-A164) mice; *Vegfr2flox/flox;Vegfr1-*

*CreERT2*: $n = 4$ (Control), 4 (VEGF-A164) mice). ****$p < 0.0001$; One-way ANOVA followed by Tukey's multiple comparison test. **e** Overexpression of VEGF in the ear of a mouse lacking *Vegfr2* in BECs and treated with the VEGF-C trap, showing only small vascular lesions (arrowheads) due to expansion of residual unrecombined cells, and no lymphatic vessel alterations, quantified in (**d**). Similar results were observed in 4/4 mice with confirmed VEGF overexpression. Scale bar: 200 μm (**b,e**, overviews), 50 μm (**b, e**, high magnification, **c**). Illustrations in (**a, e**) created using BioRender (https://www.biorender.com).

escaped recombination of the *Vegfr2flox* allele suggests their competitive advantage in the sprouting of the superficial lymphatic capillary plexus, reminiscent of the selection of LECs retaining VEGFR3 expression upon its conditional deletion during the same developmental process[9]. Strikingly, even mice that received tamoxifen during the entire period of lymphatic vascular growth and remodelling showed LYVE1+ terminal lymphatic capillary segments composed entirely of VEGFR2+ cells (Fig. 2d, f). Although some mutant mice showed a more chaotic vascular branching pattern, the overall

network complexity as measured by the number of vessel ends (Fig. 2g) and branch points (Fig. 2h) remained unchanged.

## VEGFR2 signalling level dictates LEC sprouting potential

To overcome the limitation of an independent reporter allele of Cre activity for tracking *Vegfr2*-deleted LECs, we utilized the inducible dual reporter-Cre allele *iSuRe-Cre*[26]. In combination with the *Prox1-CreERT2*, *iSuRe-Cre* allows inducible LEC-specific expression of a membrane-localized fluorescent MbTomato reporter (Tom) coupled to a

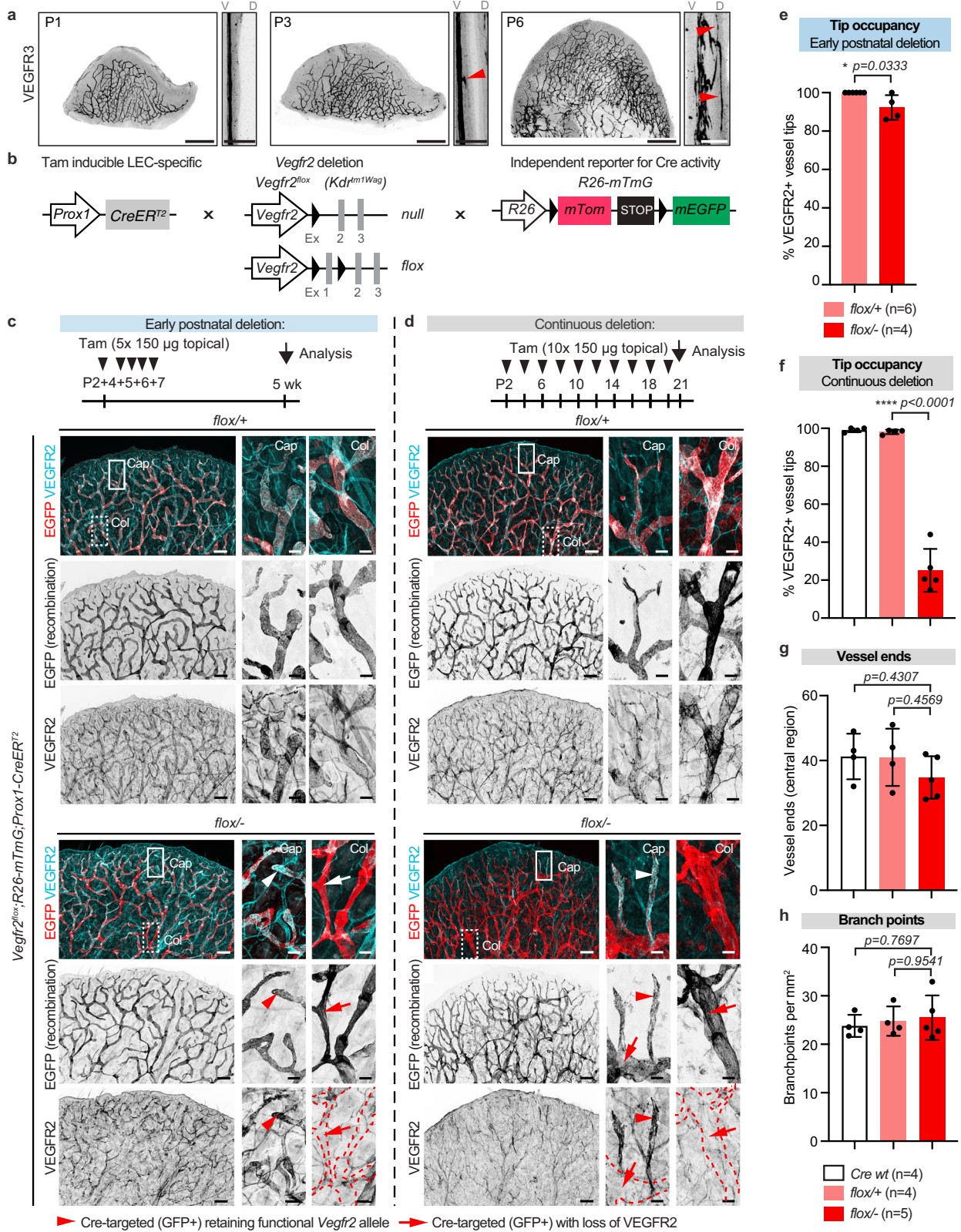

Cre-targeted (GFP+) retaining functional *Vegfr2* allele　▶ Cre-targeted (GFP+) with loss of VEGFR2

constitutively active and permanently expressed Cre recombinase, leading to faithful tracing of gene-deleted cells (Fig. 3a). The distribution of Tom⁺ *Vegfr2*-deleted LECs was analyzed in the mature ear vasculature after administration of 4-hydroxytamoxifen (4-OHT) to neonatal mice at P1 and P2 (Fig. 3b, Supplementary Fig. 2a, b).

Whole-mount immunofluorescence of the ear skin of P21 mice showed a morphologically normal network of lymphatic capillaries

and collecting vessels in *Vegfr2* mutant mice (*flox/flox*) similar to control littermates (wild type or *flox/+*) (Fig. 3c, Supplementary Fig. 2c). Recombined Tom⁺ LECs were detected in both the distal lymphatic capillaries and centrally located collecting vessels in control mice (Fig. 3c,d, Supplementary Fig. 2c). In contrast, in homozygous *Vegfr2^flox/flox* mutant mice Tom⁺ LECs were restricted to collecting lymphatic vessel network already present at P1 (Figs. 2a, 3c,d,

**Fig. 2 | Postnatal genetic deletion of *Vegfr2* leads to selection of LECs retaining VEGFR2 expression in dermal lymphatic capillaries. a** Whole mount immunofluorescence of postnatal ears showing the development of dermal lymphatic capillary plexus in the dorsal ear skin. Note the presence of a primary plexus at birth with first capillary sprouts appearing at P3 (arrow) and extending to form a superficial plexus at P6 (arrows). V, ventral side; D, dorsal side. **b** Genetic constructs for LEC-specific *Vegfr2* deletion and tracing. **c** Experimental scheme for assessing the effect of early postnatal deletion of *Vegfr2* (top) on dermal lymphatic vasculature of the ear, analyzed by whole mount immunofluorescence at 5 weeks of age (below). GFP expression indicates efficient Cre recombination in both *Vegfr2*-deficient (*flox/-*) and heterozygous control (*flox/+*) mice, and no vascular phenotype in the mutant. Note efficient depletion of VEGFR2 protein in the collecting vessels (Col, arrows), but not in the distal capillaries (Cap, arrowheads) in mutant mice. **d** Experimental scheme for assessing the effect of continuous postnatal deletion of

*Vegfr2* (top) on dermal lymphatic vasculature of the ear, analyzed by whole mount immunofluorescence at 3 weeks of age (below). Note that the extended tamoxifen administration regime is not sufficient to lead to efficient depletion of VEGFR2 protein in all distal capillaries (Cap, arrowheads), unlike in the collecting vessels (Col, arrows), in mutant mice. Boxed areas in (**c**, **d**) are magnified. Dotted lines in VEGFR2 staining in (**c**, **d**) indicate the outline of collecting vessels. **e, f** Proportion of VEGFR2+ lymphatic capillary ends in ears after early (**e**) or continuous (**f**) postnatal deletion of *Vegfr2*, represented as mean ± s.d. (*n* = 4-6 mice per genotype as indicated below **h**). Number of terminal lymphatic capillary ends (**g**) and branch points (**h**) in ears after continuous postnatal *Vegfr2* deletion, represented as mean ± s.d. (n = 4-5 mice per genotype as indicated). In (**e**), *\*p* < 0.05; Two-sided Mann-Whitney U test, (**f**–**h**), \*\*\*\**p* < 0.0001; One-way ANOVA followed by Tukey's multiple comparison test. Scale bar: 500 μm (**a**), 250 μm (**a**, cross sections; **c**, **d**).

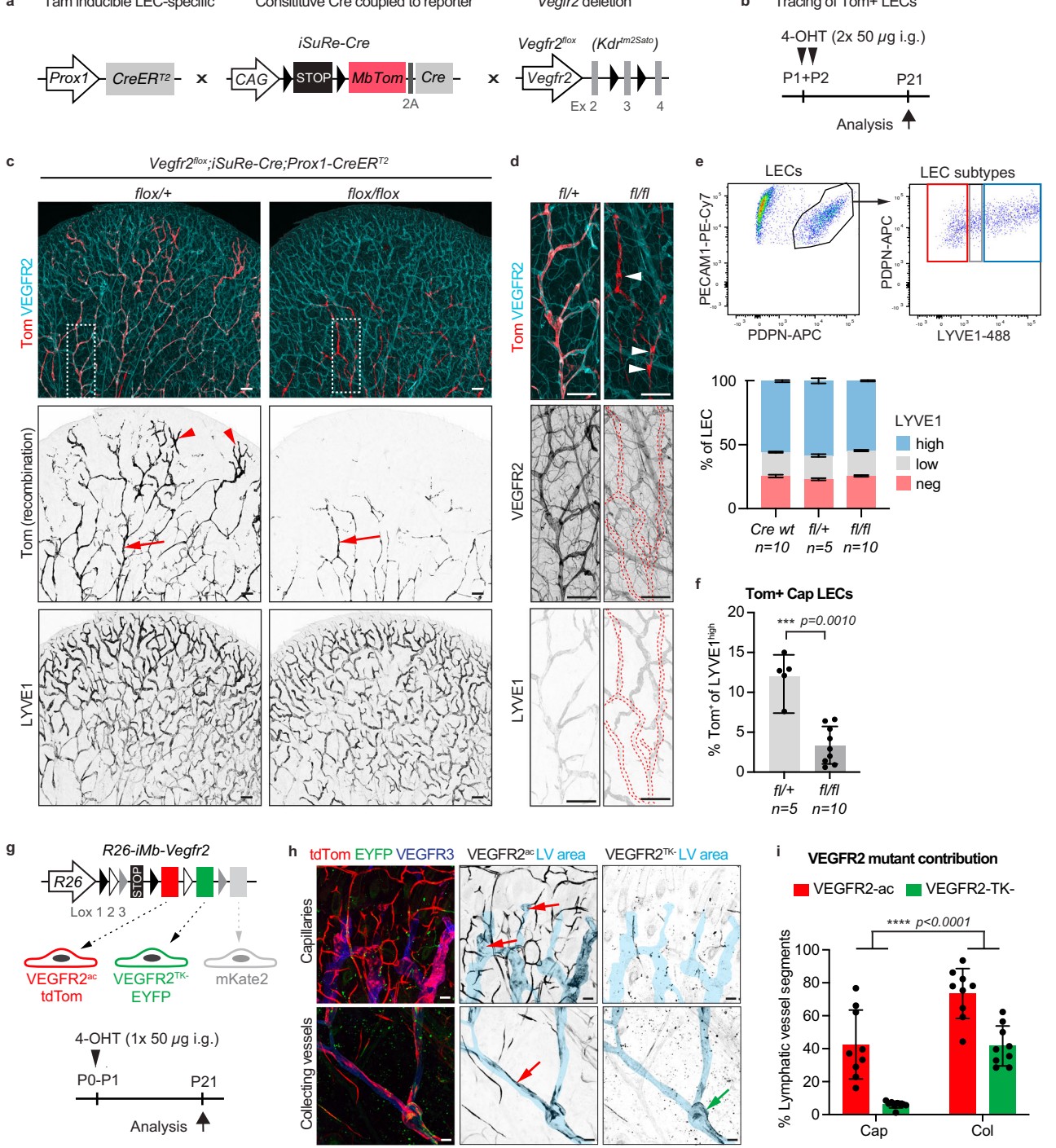

**Fig. 3 | Reduced contribution of VEGFR2 signalling-defective LECs to developing dermal lymphatic capillaries.** Genetic constructs (**a**) and experimental scheme (**b**) for tracking of *Vegfr2*-deleted LECs using the *iSuRe-Cre* allele that switches upon conditional CreER^{T2} activation to expression of constitutive Cre and MbTomato. **c, d** Whole mount immunofluorescence showing distribution of MbTom^+ LECs, labelled at early postnatal development, within mature dermal vasculature in 3-week-old mice (**c**). Cre^+ mice carrying a heterozygous (*flox/+*) *Vegfr2* allele show contribution of MbTom^+ LECs in both lymphatic capillaries (arrowhead) and (pre-)collecting vessels (arrow), *Vegfr2*-deleted (*flox/flox*) mice show labelling predominantly in (pre-)collecting vessels (arrow). Images (**c, d**) show representative data from 3 independent experiments. Boxed areas are magnified in (**d**), showing efficient VEGFR2 depletion in *Vegfr2*-deleted (*flox/flox*) mice. Dotted lines in VEGFR2 and LYVE1 staining in (**d**) indicate the outline of collecting vessels. Arrowheads in (**d**) indicate lymphatic valves. Flow cytometry analysis of LEC subtypes (**e**) and MbTom^+ LECs (**f**) in P21 ear skin. Data represent mean ± s.d. (*n* = 5-10 mice per genotype as indicated). ***p < 0.001; Two-sided Mann-Whitney U test. **g** Genetic constructs and experimental scheme for tracing ECs with different levels

of VEGFR2 activity after inducing mosaic expression of constitutively active VEGFR2 (*tdTom-Vegfr2^{ac}*) or a dominant negative tyrosine kinase mutant form (Y1173) of VEGFR2 coupled to EYFP expression (*EYFP-Vegfr2^{TK-}*). mKate2 expression is not coupled to VEGFR2 expression and is not visualized. **h** Whole mount immunofluorescence staining of P21 ear dermis of *R26-iMb-Vegfr2;Cdh5-CreER^{T2}* mice showing abundant presence of LECs expressing *tdTom-Vegfr2^{ac}* in lymphatic capillaries, while LECs expressing *EYFP-Vegfr2^{TK-}* are only found within collecting vessels. VEGFR3^+ vessels are pseudo-coloured in light blue in single channel images. Red arrows indicate LECs expressing *tdTom-Vegfr2^{ac}* and green arrow LECs expressing *EYFP-Vegfr2^{TK-}*. **i** Contribution of *EYFP-Vegfr2^{TK-}* and *tdTom-Vegfr2^{ac}* LECs to dermal lymphatic vasculature, shown as the proportion of vessel segments between two valves within LYVE1^+ capillary- (Cap), or collecting vessels (Col), with reporter-positive cells. Data represent mean ± s.d. (*n* = 450 (Cap, 5 mice) and *n* = 636 (Col, 5 mice) vessel segments; each dot represents data from one ear). ****p < 0.0001; Fisher's exact (two-tailed). Scale bar: 250 μm (**c**), 200 μm (**h**), 50 μm (**d**).

Supplementary Fig. 2c). Immunofluorescence staining confirmed that Tom^+ LECs had lost VEGFR2 expression (Fig. 3d). Although (pre-)collecting vessels formed of Tom^+VEGFR2^- LECs appeared thinner, they formed morphologically normal valves (Fig. 3d) and downregulated the lymphatic capillary marker LYVE1 (Fig. 3c), suggesting normal maturation. Flow cytometry analysis of dermal cell suspensions additionally showed similar proportions of the different populations of LYVE1^{neg}, LYVE1^{low} and LYVE1^{high} LECs, representing collecting, pre-collecting and capillary vessels, respectively, in both mutant and control mice (Fig. 3e), confirming the presence of a normal hierarchical lymphatic network in the mutants. However, in agreement with the immunofluorescence data (Fig. 3c), Tom^+ LECs were underrepresented in LYVE1^{high} capillary LEC population in *Vegfr2*-deleted mutant mice in comparison to the control mice (Fig. 3f).

To further investigate the role of VEGFR2 signalling in developmental lymphangiogenesis, we utilized the *R26-iMb-Vegfr2* mosaic mice[27] crossed to the pan-endothelial *Cdh5-CreER^{T2}* line[28] (Fig. 3g). In these mice, stochastic recombination of the *iMb-Vegfr2* allele results in mutually exclusive expression of either a constitutively active form of VEGFR2 without the extracellular domain[29] coupled to tdTomato expression (*tdTom-Vegfr2^{ac}*), or a dominant negative tyrosine kinase mutant form (Y1173) of VEGFR2 coupled to EYFP expression (*EYFP-Vegfr2^{TK-}*). Tracing of ECs after 4OHT induction at P1 revealed the abundant *tdTom-Vegfr2^{ac}* LECs throughout the dermal lymphatic vasculature, including lymphatic capillary tips (Fig. 3h). In contrast, LECs expressing the signalling-defective VEGFR2 (*EYFP-Vegfr2^{TK-}*) were excluded from lymphatic capillaries but persisted in collecting vessels at the base of the ear, including valve regions (Fig. 3h). Quantifying lymphatic vessel segments containing reporter-expressing cells confirmed the contribution of *tdTom-Vegfr2^{ac}* LECs to both vessel compartments, while *EYFP-Vegfr2^{TK-}* LECs were largely absent from capillaries (Fig. 3i).

Together, these results show that LECs with impaired VEGFR2 signalling lack the competency to contribute to superficial lymphatic capillary formation and instead remain confined to the collecting vessel network present at the time of neonatal induction.

### Efficient VEGFR2 loss impairs developmental lymphangiogenesis

A caveat of the *iSuRe-Cre* line is its relatively low sensitivity towards initial CreER activity. While true mutant cells are faithfully reported, the low recombination efficiency may result in cells loosing VEGFR2 expression without recombining the *iSuRe-Cre* allele, leading to false negatives[26]. To overcome this, we crossed the improved *R26-iSuRe-HadCre* allele[11] to *Vegfr2^{flox};Prox1-CreER^{T2}* mice (Fig. 4a), and induced gene deletion either prior to lymphatic capillary sprouting in neonatal mice (P1-P2) or continuously during the entire period of ear vascular

growth (P2-P20) (Fig. 4b). While *iSuRe-Cre* constitutively expresses Cre-recombinase, *R26-iSuRe-HadCre* only transiently amplifies Cre activity, after which it self-recombines and replaces the Cre cassette with a tdTomato reporter gene (Fig. 4a). Moreover, the *R26-iSuRe-HadCre* allele is more sensitive to the initial CreER recombination, increasing the fraction of true positive mutant cells analyzed, as illustrated by near complete recombination of the dermal ear vasculature in control mice (Fig. 4c) using the neonatal induction scheme that resulted in mosaic labelling in *iSuRe-Cre* mice (Fig. 3c). However, similar to observations from the *iSuRe-Cre* mice, Tom^+ LECs that lost VEGFR2 expression were unable to contribute to the postnatally developing dermal lymphatic capillary bed and were only found in collecting lymphatic vessels (Fig. 4c, d). In these ears, unrecombined, VEGFR2-expressing LECs outcompeted VEGFR2-deficient cells, resulting in the formation of a morphologically normal lymphatic capillary plexus in the distal margin of the tissue (Fig. 4c). To overcome this compensatory effect of wild type LECs, we induced recombination every second day from P2 to P20 (Fig. 4b), ensuring that even LECs escaping initial recombination would be targeted by subsequent tamoxifen administration during the entire critical neonatal period of vessel growth. Whole-mount analysis of dermal ear skin revealed a hypoplastic lymphatic vascular network with a poorly developed capillary plexus in VEGFR2-deficient mice compared to control mice (Fig. 4e,f), characterized by reduced vessel coverage, branching and capillary ends (Fig. 4g-i). Strikingly, even this extensive tamoxifen regime resulted in some capillary ends of mutant mice composed of non-recombined VEGFR2^+ LECs (Fig. 4f).

Neonatal VEGF-C inhibition via systemic ligand trapping leads to a complete loss of dermal lymphatic vessel network[9,20] (Supplementary Fig. 3a), while adult inhibition has no effect in most organs[20]. Previous attempts to simultaneously genetically delete the two receptors during development have not recapitulated the VEGF-C-trap phenotype[9,10], likely due to the selection of LECs that escaped recombination, as discussed above. Using the *R26-iSuRe-HadCre;Prox1-CreER^{T2}* alleles, we found that efficient homozygous deletion of both *Vegfr2* and *Vegfr3* in LECs in neonatal mice resulted in the complete loss of the dermal lymphatic network in the ear dermis at three weeks of age (Fig. 4j, Supplementary Fig. 3a). A single *Vegfr2* allele was sufficient to rescue LEC survival, but not sprouting (Fig. 4j, Supplementary Fig. 3a). Although double deletion of *Vegfr2* and *Vegfr3* resulted in efficient depletion of the receptors at the protein level, abnormal vessel networks formed in some mice from LECs that escaped *R26-iSuRe-HadCre* recombination and retained VEGFR2/3 expression (Supplementary Fig. 3b). Similarly, in mice with *Vegfr3* deletion alone, Cre-targeted tdTom^+ LECs survived, but ectopic sprouts formed from tdTom^- non-recombined cells (Supplementary Fig. 3b). Interestingly, some of

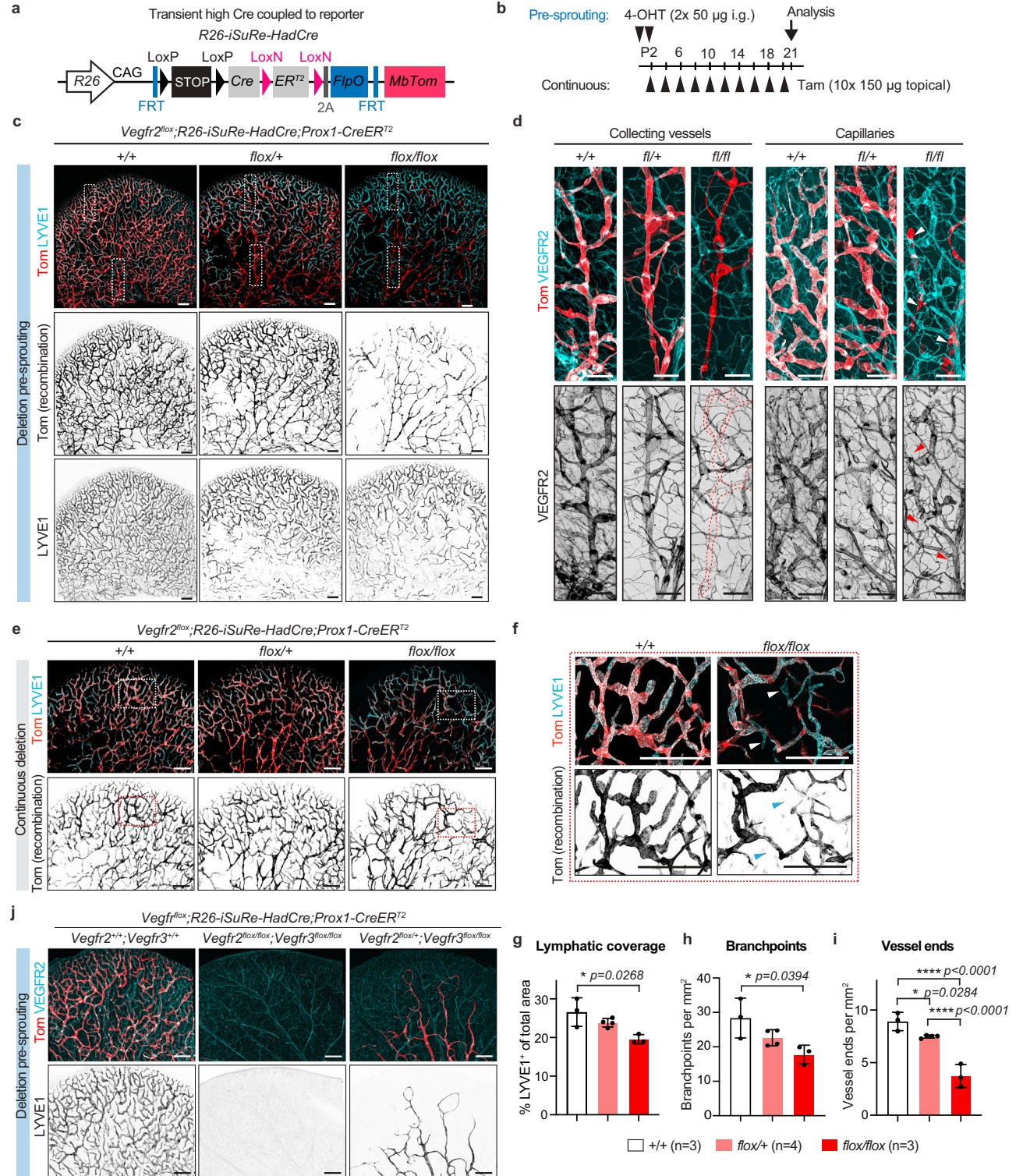

these mice additionally showed abnormally patterned tdTom+ vessels with low levels of VEGFR3 (Supplementary Fig. 3b), suggesting incomplete recombination of only one *Vegfr3^flox* allele. In agreement with previous findings[9], combined deletion of *Vegfr2* and *Vegfr3* in adult mice at 10 weeks of age did not cause apparent lymphatic defects five weeks after gene deletion (Supplementary Fig. 3c).

Together, these results show that LECs with defective VEGFR2 signalling are outcompeted by those retaining VEGFR2 expression during the formation of superficial lymphatic capillaries.

This compensation results in the formation of a phenotypically normal lymphatic network, explaining why the VEGFR2-deficient phenotype has not been previously reported. Only continuous targeting using a high-fidelity genetic approach and throughout active vessel growth led to capillary hypoplasia, demonstrating that VEGFR2 is essential for developmental lymphatic capillary sprouting, a role that requires but cannot be compensated by VEGFR3 in the tested models. Within this developmental context, both VEGFR3 and VEGFR2 are also needed for LEC survival.

**Fig. 4 | Impaired developmental lymphangiogenesis upon efficient deletion of *Vegfr2* in LECs.** Genetic constructs (**a**) and experimental scheme (**b**) for efficient deletion and faithful tracing of *Vegfr2* or double *Vegfr2/Vegfr3*-deleted LECs using *R26-iSuRe-HadCre*. **c**, **d** Whole mount immunofluorescence showing the distribution of MbTom⁺ LECs, labelled at early postnatal development (P1, P2), within the mature dermal vasculature in 3-week-old mice. Cre⁺ mice carrying only *R26-iSuRe-HadCre* (+/+) or heterozygous (*flox/+*) *Vegfr2* allele show contribution of MbTom⁺ LECs in both lymphatic capillaries and collecting vessels, while *Vegfr2*-deleted (*flox/flox*) mice show labelling predominantly in collecting vessels. Boxed areas in (**c**) are magnified in (**d**) to show limited contribution of recombined cell to lymphatic capillaries and efficient deletion of VEGFR2 in Tom⁺ collecting lymphatic vessels in *Vegfr2^flox/flox^* mice. In (**d**), dotted lines in VEGFR2 staining indicate the outline of collecting vessels, and arrowheads point to non-recombined LECs in mutant mice. Images in (**c**, **d**) represent data from two independent experiments. Whole mount immunofluorescence (**e** and magnification of boxed areas in (**f**)) and quantification of vessel area (**g**), branchpoints (**h**) and terminal lymphatic capillary ends (**i**), showing that continuous deletion of *Vegfr2* results in hypoplastic lymphatic network in *Vegfr2*-deleted (*flox/flox*) mice compared to controls. Arrowheads in (**f**) point to non-recombined LECs at vessel tips in mutant mice. Data in (**g–i**) represent mean ± s.d. (*n* = 3–4 mice per genotype as indicated). *$p < 0.05$, ****$p < 0.0001$; One-way ANOVA followed by Tukey's multiple comparison test. **j** Whole mount immunofluorescence of P21 ear dermis showing the effect of neonatal high-fidelity deletion of *Vegfr2* and/or *Vegfr3*. Images in (**j**) represent data from two independent experiments. Scale bar: 500 μm (**c**, **e**, **f**, **j**), 200 μm (**d**).

## VEGFR2 is required for VEGF-C–VEGFR3–PI3Kα-mediated sprouting

To assess the potential function of VEGFR2 in the mature vasculature, we analyzed VEGF-C-induced lymphangiogenic responses following LEC-specific deletion of *Vegfr2* (in combination with the *iSuRe-Cre* reporter) and compared them to responses in mice with deletion of *Vegfr3* (in combination with the *R26-mTmG* reporter) (Fig. 5a). Given that in vitro studies have suggested a specific requirement for VEGFR2 in VEGF-C–VEGFR3-induced PI3K-AKT activation[14] we further compared these effects to those observed upon deletion of *Pik3ca*, which encodes the downstream effector PI3Kα. To circumvent selection of unrecombined LECs occurring during developmental vessel growth (see above and ref. [9]), we induced gene deletion in the quiescent fully formed mature vasculature, where receptor deletion does not lead to developmental selection. One (*Vegfr3^flox^*) or three consecutive (*Vegfr2^flox^*, *Pik3ca^flox^*) administrations of tamoxifen, adjusted to reflect the different recombination efficiencies of the respective alleles, were used to achieve complete gene deletion (Fig. 5a). Two weeks later, the mice additionally received an intradermal injection of an AAV vector encoding full length VEGF-C protein or its fully processed ΔNΔC form into the ear skin (Fig. 5a).

Whole-mount immunofluorescence of the ear skin of *Vegfr3^flox^* mice showed efficient Cre-mediated recombination as indicated by GFP reporter expression and loss of VEGFR3 (Fig. 5b). The VEGF-C-induced lymphangiogenic response observed in wild type (Fig. 1b) and heterozygous controls was completely abrogated by homozygous deletion of *Vegfr3* (Fig. 5b). This illustrates the requirement of VEGFR3 for sprouting lymphangiogenesis and confirms that sole activation of VEGFR2 (Fig. 1d) does not elicit this response. Importantly, this effect was independent of the processing of VEGF-C as AAV-VEGF-CΔNΔC encoding the mature fully processed form capable of activating VEGFR2, similarly did not result in a lymphangiogenic response in mice deficient for *Vegfr3* specifically (Fig. 5b). *Pik3ca* deletion similarly completely abrogated VEGF-C-induced lymphatic hyperplasia, while control littermates showed the expected hypersprouting response (Fig. 5c,d). Surprisingly, however, genetic deletion of *Vegfr2* led to a reduced lymphangiogenic response that was uniquely characterized by vessel hyperplasia and lack of sprouting (Fig. 5e), recapitulating the response caused by selective activation of VEGFR3 (Fig. 1b). While *Pik3ca* deletion led to reduced VEGFR3 levels (Fig. 5c and ref. [30]), this was not observed after *Vegfr2* deletion (Fig. 5e).

Quantitative flow cytometric analysis of dermal LECs from control or heterozygous *Vegfr2/3* mice confirmed immunofluorescence data of VEGF-C-induced LEC proliferation, as indicated by an increase in the LEC/EC ratio and associated increase in the KI67⁺ LEC population two weeks after the treatment (Fig. 5f, g). However, no increase in the proportion of KI67⁺ LECs in reporter-expressing Cre-targeted LECs was observed in *Vegfr3*-deleted mice, indicative of the lack of expansion of the LEC population (Fig. 5f). In contrast, Cre-targeted Tom⁺ i.e. *Vegfr2*-deleted LECs showed similar proliferation and only a modest reduction

in the expansion of the LEC population (Fig. 5g) compared to heterozygous *Vegfr2* or wild-type controls, respectively.

Early time-point analysis of AAV-VEGF-C-treated ears revealed rapid induction of lymphatic sprouting together with robust LEC proliferation, which was detectable not only at capillary sprouts but also in morphologically quiescent, blunt-ended capillaries and other vessel segments (Fig. 5h, Supplementary Fig. 4a, b). In contrast, LEC-specific activation of PI3K signalling by *Vegfr3-CreER^T2^*-mediated overexpression of a constitutively active PI3Kα, encoded by *Pik3ca^H1047R^*, promoted robust sprouting, in line with previous observations[18], with proliferating LECs initially largely restricted to vessel sprouts (Fig. 5i, Supplementary Fig. 4a, b).

These results demonstrate that VEGFR3 activation is sufficient for VEGF-C-induced LEC proliferation, which can precede sprouting, whereas concurrent activation of VEGFR2 is required for VEGF-C–VEGFR3–PI3Kα-mediated lymphatic vessel sprouting.

## VEGFR2 is required for regenerative lymphangiogenesis

To assess the functional relevance of VEGFR2 in a disease-relevant context, we examined lymphatic vessel regrowth following ear skin injury induced by a punch biopsy, used to model regenerative lymphangiogenesis in back[31] and ear skin[32,33]. Regenerative lymphangiogenesis was analyzed in mice with LEC-specific deletion of *Vegfr2*, *Vegfr3*, or both receptors (Fig. 6a, Supplementary Fig. 5). Quantitative analysis revealed a significant reduction in lymphatic sprouting, assessed by the abundance of spiky vessel ends, at the wound edge in *Vegfr2*-deficient mice, which was further suppressed upon loss of *Vegfr3* (Fig. 6b, Supplementary Fig. 5). In contrast, the total lymphatic vessel area formed in the wound region over the one-week observation period was modestly reduced in *Vegfr2*-deficient mice compared to controls, whereas it was strongly decreased in *Vegfr3*-deficient mice (Fig. 6b, c). These findings indicate that while VEGFR3 is indispensable for lymphatic regeneration, VEGFR2 plays a selective and critical role in regenerative lymphatic sprouting rather than vessel expansion, consistent with its essential function in VEGF-C–mediated lymphatic sprouting.

## VEGFR2 is highly expressed and activated on LEC surface

To further clarify the receptor-specific cellular functions, we analyzed VEGFR2 and VEGFR3 expression, subcellular localization and activation status in vivo. Single cell RNA sequencing of dermal LECs from wild type adult mouse ear skin[18] showed higher expression of *Flt4/Vegfr3* and *Nrp2* in LECs of lymphatic capillaries compared to those of collecting lymphatic vessels and valves, while *Vegfr2/Kdr* expression was similar in all lymphatic vessel subtypes (Fig. 7a). *Nrp1*, implicated in the regulation of VEGFR3 signalling in vitro[14], was not expressed in adult murine dermal LECs in vivo indicating functional differences in in vitro vs in vivo mechanisms (Fig. 7a).

Whole-mount immunofluorescence confirmed the expression of VEGFR2 protein in all endothelia, and VEGFR3 expression specifically in LECs, in the mature dermal vasculature of a 4-week-old wild-type

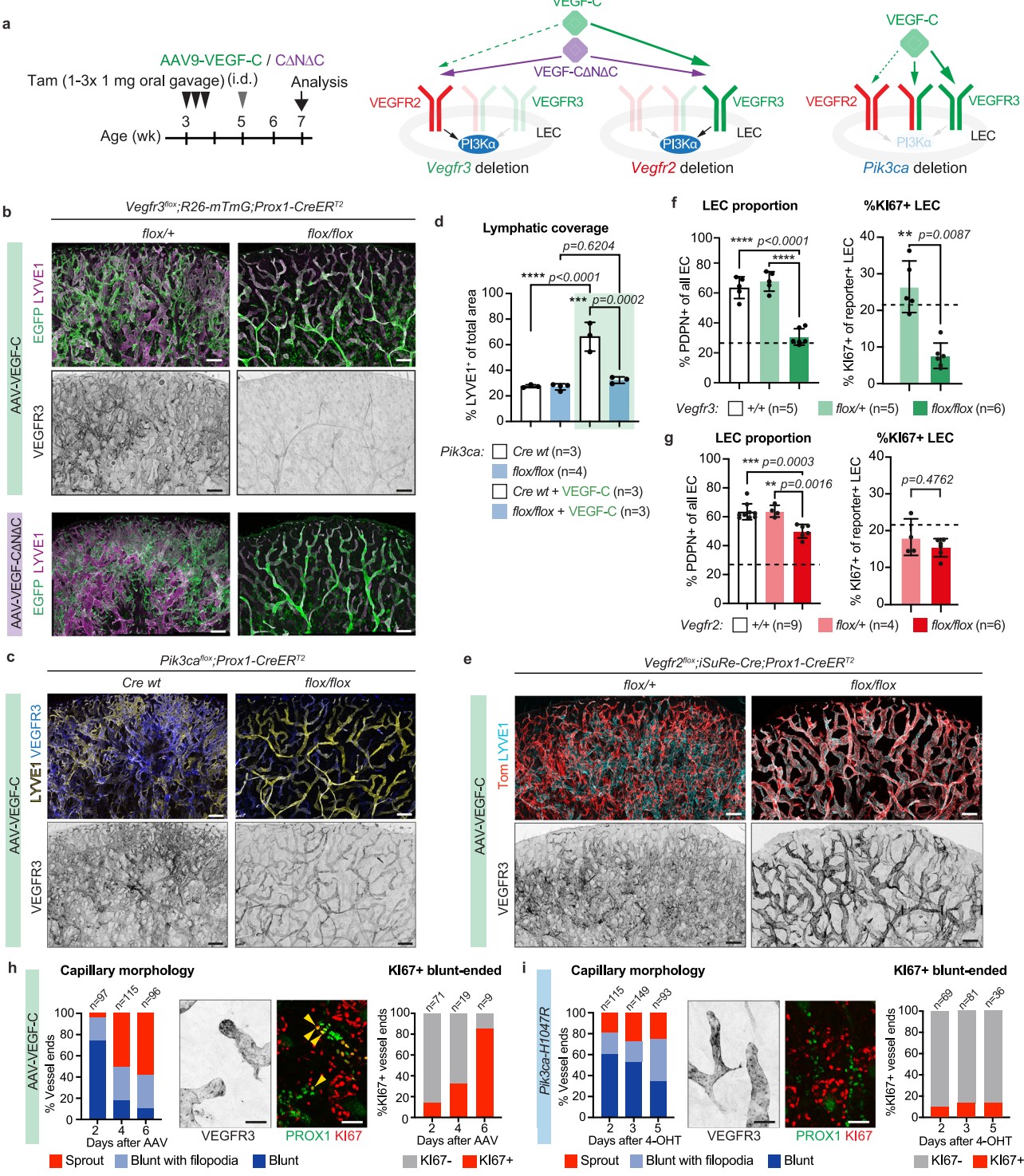

mouse ear (Fig. 7b). To assess cell-surface receptor localization, tissues were stained under non-permeabilized conditions, which was previously used to evaluate VEGFR3 distribution[30]. Control stainings for intracellular markers showed negligible tubulin and phalloidin signal in unpermeabilized tissue (Supplementary Fig. 6a), indicating limited intracellular access. PROX1 staining was absent in most LECs, with only a weak signal detected in a small subset, likely due to partial membrane disruption. Consistent differences in VEGFR2 and VEGFR3 patterns between permeabilized and unpermeabilized conditions further support the validity of this approach for assessing receptor localization in vivo (Fig. 7b). While immunostaining of unpermeabilized tissue showed that cell-surface VEGFR3 was detected in LECs only, we

unexpectedly observed a more abundant presence of cell-surface VEGFR2 in LECs compared to BECs (Fig. 7b). A similar pattern was observed in the developing dermal vasculature of the back skin at embryonic day (E) 15, showing increased relative abundance of VEGFR2 on the surface of LECs in comparison to BECs (Fig. 7c). At this embryonic stage VEGFR3 was expressed in both LECs and BECs but remained at low levels and largely absent from the surface of BECs (Fig. 7c).

Ligand-induced activation of VEGFR2 and VEGFR3 has been shown to lead to their internalization through endocytosis, which is required for downstream signalling[14,34,35]. Elevated intracellular receptor levels and perinuclear or Golgi-associated localization have been

**Fig. 5 | Distinct requirements for VEGFR2 and VEGFR3 in VEGF-C-induced lymphangiogenesis. a** Experimental scheme and ligand-receptor interactions for assessing VEGF-C-induced lymphangiogenic response after genetic deletion of *Vegfr2*, *Vegfr3* or *Pik3ca* in the mature lymphatic vasculature. Two weeks after induction mice received an intradermal (i.d.) injection of AAV-VEGF-C into the ear skin and were analyzed 2 weeks later. VEGF-CΔNΔC, fully processed mature form of VEGF-C capable of binding VEGFR2. VEGF-C-induced lymphangiogenic response in mice lacking lymphatic endothelial *Vegfr3* (**b**), *Pik3c*a (**c**, quantified in **d**) or *Vegfr2* (**e**), and their respective littermate controls. Note loss of lymphatic hyperplasia in *Vegfr3*-deleted (**b**, 1 mg of Tam) and *Pik3ca*-deleted (**c**, 3 ×1mg of Tam) mice and reduced sprouting in *Vegfr2*-deleted mice (**e**, 3 × 1 mg of Tam). Data in (**d**) represent mean vessel area (*n* = 3–4 mice per genotype as indicated) ± s.d., ***p < 0.001 and ****p < 0.0001, Two-sided Mann-Whitney U test. Images in (**b**, **e**) represent data

from 3 independent experiments. Flow cytometry analysis of LEC proliferation after VEGF-C stimulation, showing reduced proliferation (KI67⁺) and expansion of the total LEC population in *Vegfr3*-deleted mice (**f**), while *Vegfr2*-deleted LECs show similar proliferation and only a modest reduction in expansion of LEC population (**g**) compared to controls. Data represent mean % of LECs (*n* = 4-9 mice per genotype as indicated) ± s.d. Dotted lines indicate %KI67⁺ LECs and proportion of LECs of all ECs in untreated wild type (+/+) mice (**f**, **g**). **p < 0.01, ***p < 0.001 and ****p < 0.0001; Two-sided Mann Whitney U test. Early time-point analysis of lymphatic capillary morphology and LEC proliferation (KI67 + PROX1 + , yellow arrows) in AAV-VEGF-C-treated mice (**h**) and in *Pik3ca^H1047R*;*Vegfr3-CreER^T2* mice following topical application of 4-OHT to activate PI3Kα (**i**). 3-5 mice were analysed for each time point, and the number of vessel ends analyzed is indicated. Scale bar: 250 μm (**b**, **c**, **e**), 50 μm (**h**, **i**).

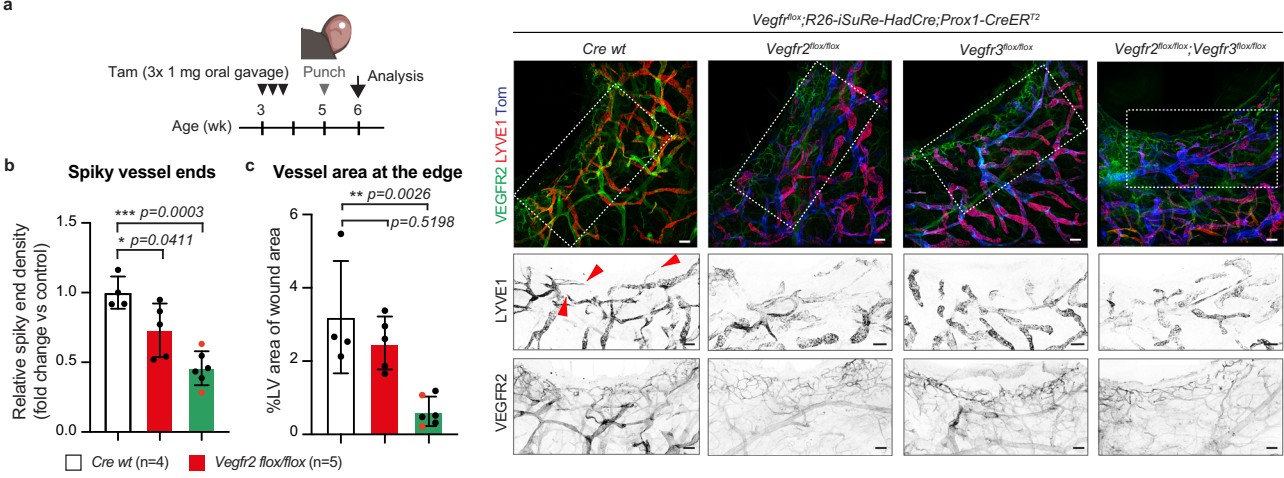

**Fig. 6 | VEGFR2 promotes lymphatic sprouting during regenerative lymphangiogenesis. a–c** Experimental scheme and whole mount immunofluorescence of the punched ear skin. Boxed areas are magnified below. Arrowheads in LYVE1 staining indicate the spiky vessel ends on the edge of the wound area. Quantification of the spiky vessel ends (**b**) and vessel area (**c**) at the wound edge showing the vessel regrowth during one-week period after punching

was reduced in *Vegfr2*-deficient mice and further suppressed upon loss of *Vegfr3*. Data in (**b**, **c**) represent mean ± s.d. (*n* = 4–6 mice per group as indicated), normalized to control values. *Vegfr3*-deleted and *Vegfr2/Vegfr3* double-deleted (red dots) mice were combined into one group. *p < 0.05, **p < 0.01, ***p < 0.001; One-way ANOVA followed by Tukey's multiple comparison test. Scale bar: 100 μm (**a**). Illustration in (**a**) created using BioRender (https://www.biorender.com).

used as indicators of active VEGFR signalling in vivo[28,30,36], with high cell-surface abundance instead potentially reflecting reduced signalling activity. Based on these observations, we assessed VEGFR2 and VEGFR3 localization and abundance following AAV-VEGF-C administration at 5 weeks of age, with analysis performed two weeks later. As expected, VEGF-C overexpression induced pronounced perinuclear accumulation of VEGFR3 (Fig. 7d, e), consistent with active signalling and receptor internalization. Strikingly, in contrast to VEGFR3, VEGF-C stimulation led to a marked increase in VEGFR2 protein levels in LECs, detectable in both permeabilized and non-permeabilized tissue (Fig. 7d–f), but not in BECs (Supplementary Fig. 7). Consistent with the previously reported role of PI3K signalling in maintaining VEGFR3 cell surface expression[30,37], baseline VEGFR3 protein levels were reduced in *Pik3ca* deficient vessels, with near complete absence of detectable surface VEGFR3 in these mutants following VEGF-C stimulation (Fig. 7e). In contrast, steady-state VEGFR2 levels were not affected by *Pik3ca* loss, yet the VEGF-C induced increase in VEGFR2 was completely abolished in the mutant vessels (Fig. 7e,f). These findings indicate that PI3Kα signalling is required for VEGF-C-dependent regulation of VEGFR2 abundance and suggest that loss of PI3Kα disrupts VEGF receptor trafficking dynamics.

To directly assess the activation status of VEGFRs, we applied proximity ligation assay (PLA) using an antibody recognizing VEGFR2 or VEGFR3 in combination with a pan anti-phospho-Tyrosine (pTyr)

antibody. To localize phosphorylated receptors within intact vascular structures, we established a PLA protocol for the analysis of whole-mount embryonic skin tissue. PLA signals correlated with the cell surface staining of the receptors (Fig. 7g), indicating activation of both receptors in LECs. VEGFR2 phosphorylation was observed in both BECs and LECs, while VEGFR3-pTyr was detected predominantly in LECs (Fig. 7g). In addition, we detected proximity of VEGFR2 and VEGFR3 in LECs in vivo using the same method (Fig. 7h), suggesting the presence of receptor heterodimers in lymphangiogenic sprouts. PLA signal was confirmed to be specific as technical controls, where one antibody was omitted, showed negligible signal (Supplementary Fig. 6b). Additionally, biological controls representing embryos with mosaic LEC-specific genetic deletion of *Vegfr3* showed loss of VEGFR3-pTyr PLA signal in *Vegfr3*-deleted cells (Supplementary Fig. 6c).

Collectively, these data show that despite similar total levels, VEGFR2 protein is more abundantly present on the cell surface of LECs in comparison to BECs in both proliferative embryonic and quiescent postnatal vessels. Tyrosine phosphorylation and proximity to VEGFR3 additionally suggest active VEGFR2 signalling and heterodimer formation in lymphatic endothelia in vivo.

## Discussion
VEGF-C signalling is crucial for lymphangiogenesis and has been extensively studied in different developmental and pathological

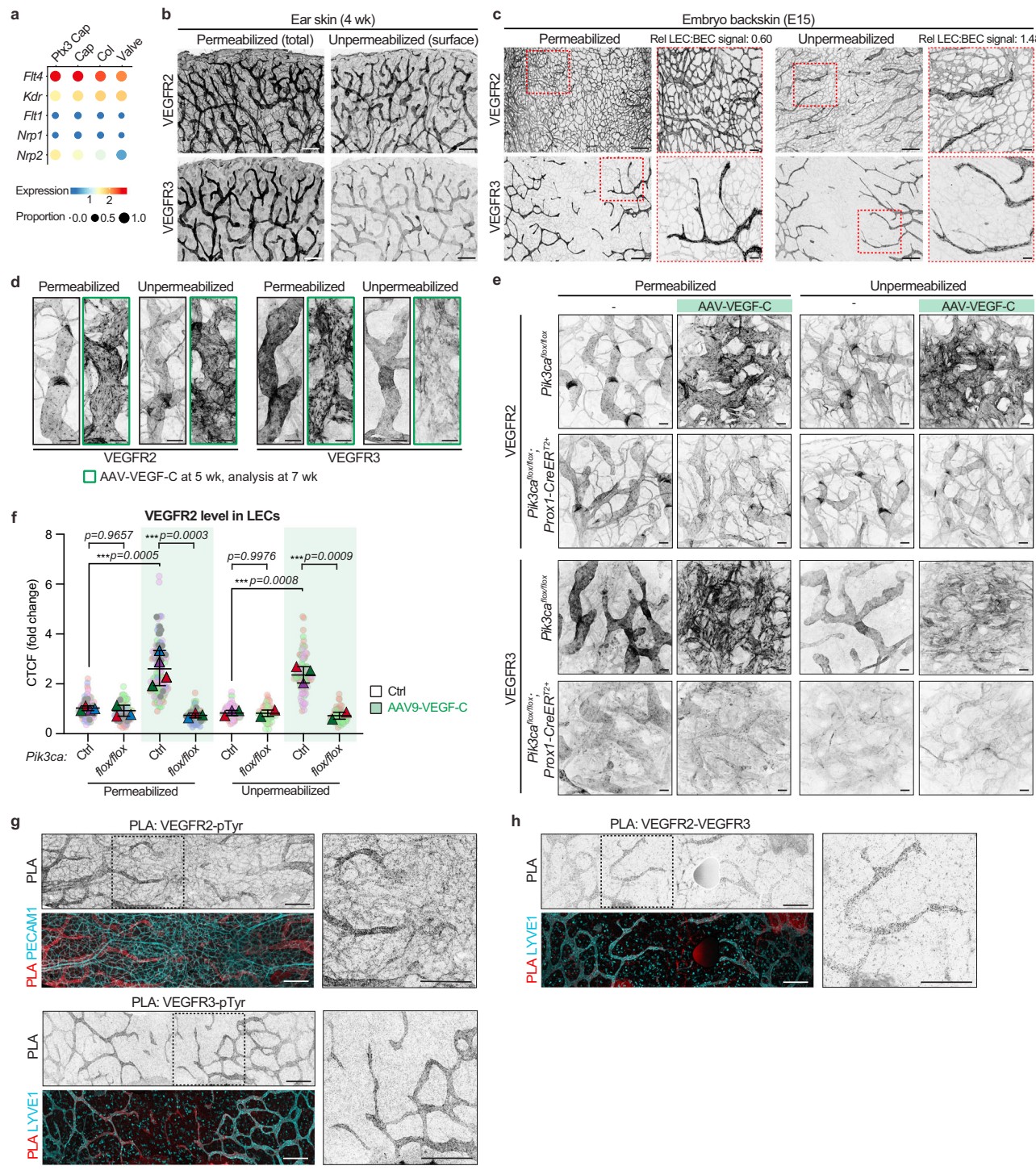

contexts. Our study confirms the previously established essential function of VEGFR3 in mediating the VEGF-C response, but unexpectedly also uncovers the exquisite requirement of VEGFR2 for VEGF-C–VEGFR3-mediated vessel sprouting during developmental and neolymphangiogenic growth. Importantly, our findings help explain why this essential function of VEGFR2 has not been identified previously. Limitations in the fidelity of genetic targeting and reporting tools, combined with the strong competitive advantage of VEGFR2+ LECs during sprouting, allow non-recombined cells to outcompete those lacking VEGFR2. As a result, these residual escaper LECs can compensate and form a morphologically normal vascular network, thereby obscuring the developmental phenotype – a phenomenon recently also observed in angiogenic blood vessels of the retina[11].

Previous loss-of-function studies investigating the role of VEGFR2 in lymphatic vessels have yielded an unclear picture. Embryonic deletion of *Vegfr2* using *Lyve1-Cre* resulted in a hypoplastic lymphatic capillary plexus, while valve formation and collecting vessel maturation appeared largely normal, supporting a role for VEGFR2 in early lymphatic vessel development[38]. However, *Lyve1-Cre* is not LEC-specific and recombines in multiple blood vascular beds, including the yolk sac, liver and lung. Loss of VEGFR2 in these tissues severely impaired vascular development, likely contributing to the reduced survival reported in *Vegfr2^{lox/lox}*; *Lyve1-Cre* embryos, most of which died by embryonic day 14.5[38]. The small fraction of surviving animals complicates the interpretation of a strictly cell-autonomous role for VEGFR2 in LECs, as confounding effects arising from blood vascular

**Fig. 7 | Expression, cell surface localization and activation of VEGFR2 and VEGFR3 and their heterodimers in lymphatic endothelium. a** ScRNA-seq bubble plots showing expression of genes encoding the VEGF/VEGF-C receptors (*Vegfr1/Flt1, Vegfr2/Kdr, Vegfr3/Flt4*) and co-receptors (*Nrp1, Nrp2*) in dermal LEC types from mouse ear skin. Data extracted from[18]. Col, Collecting vessel; (Ptx3) Cap, (terminal Ptx3[+]) lymphatic capillary. Whole mount immunofluorescence of 4-week-old mouse ear dermis (**b**), and E15 embryonic back skin (**c**) for total (permeabilized) or cell surface (unpermeabilised) levels of VEGFR2 and VEGFR3. Boxed regions in (**c**) are magnified on the right. Relative LEC:BEC mean VEGFR2 fluorescence staining intensity is indicated. Images represent data from two (**b**) or a single (**c**) experiment(s). Whole mount immunofluorescence of the ear skin showing the effect of VEGF-C (**d**, **e**) and *Prox1-CreER^{T2}*-mediated deletion of *Pik3ca* (**e**) on VEGFR2 and VEGFR3 levels and localization. Recombination was induced at 3 weeks by five consecutive administrations of tamoxifen (1 mg). Two weeks later, AAV9-VEGF-C was intradermally injected into the ear skin and mice (*n* = 2–4 per condition) were analyzed two weeks later. Ctrl, control group including Cre- mice and Cre+ not carrying the floxed allele (see Source Data). Images in (**d**) represent data from two independent experiments. **f** Quantification of VEGFR2 levels from (**e**), represented as corrected total cell fluorescence (CTCF) normalized to control values. Data are shown as a superplot: round symbols represent single measurements, triangles indicate mean value per mouse (*n* = 4 (*Cre wt* ± VEGF-C, permeabilized); *n* = 3 (*flox/flox* ± VEGF-C, permeabilized); *n* = 3 (*Cre wt* ± VEGF-C, unpermeabilized); *n* = 2 (*flox/flox* ± VEGF-C, unpermeabilized) mice, 20-30 measurements per mouse). Statistical analyses were performed on mouse means and shown as mean ± s.d. ***p < 0.001; One-way ANOVA followed by Tukey's multiple comparison test. **g** Whole mount PLA of activated VEGFR2 or VEGFR3 in E15 skin, detected using anti-phospho-Tyrosine and anti-VEGFR antibodies, co-stained with PECAM1 or LYVE1. (**h**) Whole mount PLA for the detection of VEGFR2 and VEGFR3 heterodimers in E15 skin. Images in (**g**, **h**) represent data from 3 independent experiments. Boxed regions in (**g**, **h**) are magnified on the right. Scale bar: 250 µm (**b**, **c**), 50 µm (**d**, **e**, **g**, **h**).

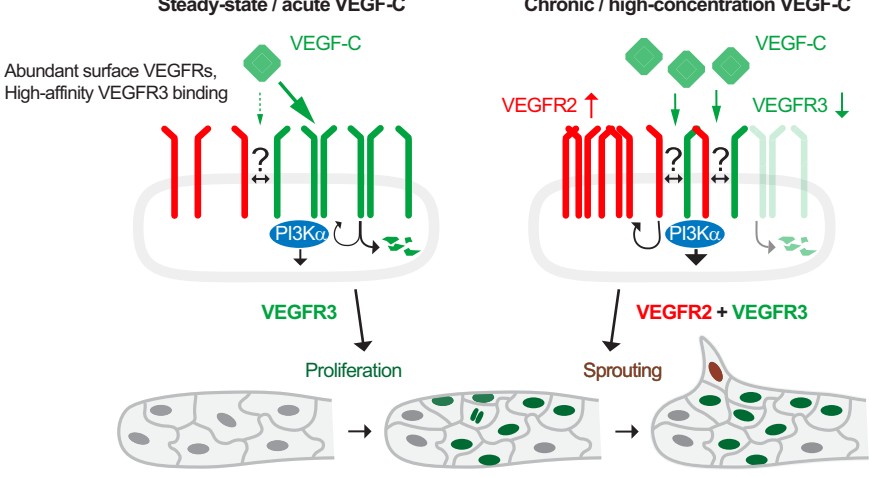

Coordinated LEC proliferation and lymphatic vessel sprouting via VEGF-C signalling

**Fig. 8 | Proposed model of VEGF-C-mediated lymphangiogenesis.** VEGF-C coordinates receptor abundance and signalling via VEGFR3 homodimers and VEGFR2/3 heterodimers. VEGF-C initially binds to its high-affinity receptor VEGFR3, inducing internalization and LEC proliferation. Concomitant VEGF-C-induced PI3Kα-dependent increase in VEGFR2 cell surface abundance shifts signalling toward VEGFR2/3 heterodimers, promoting vessel sprouting. This mechanism couples LEC proliferation with sprouting, ensuring adequate LEC numbers for functional lymphangiogenesis. While alternative modes of receptor cooperation cannot be excluded (denoted by?), functional and proximity-based data are most consistent with VEGF-C-induced VEGFR2/3 heterodimer signalling in vivo.

defects cannot be excluded. Similarly, pan-endothelial deletion of *Vegfr2* using *Cdh5-CreER^{T2}* in early postnatal or adult mice resulted in lymphatic defects in certain organs, including the trachea, but concomitant blood vascular abnormalities preclude exclusion of indirect effects also in this study[10]. Notably, LEC-specific *Vegfr2* deletion in early postnatal or juvenile mice caused no apparent lymphatic phenotypes[9], which we now show reflects the competitive replacement of *Vegfr2*-deficient LECs by unrecombined cells.

Gain-of-function approaches have likewise failed to provide a consistent understanding of VEGFR2 activity in vivo. Although specific activation of VEGFR2 signalling potently promotes proliferation, migration and survival in cultured LECs, its specific stimulation in vivo utilizing VEGF has been reported to only modestly increase lymphatic vessel diameter[4] —an unexpectedly mild effect that remains unexplained. Notably, our finding that even this modest effect was abolished in mice with BEC-specific deletion of *Vegfr2* supports previous reports suggesting that the observed in vivo response results from secondary effects mediated by activation of VEGFR2 in the blood vascular endothelium. This promotes vascular permeability and inflammation, which in turn facilitates the recruitment of VEGF-C producing macrophages[16,17]. Importantly, our results demonstrate that sole activation of VEGFR3 homodimers is insufficient to induce proper sprouting lymphangiogenesis. Specifically, while sole VEGF-C–VEGFR3

activation, achieved in the absence of VEGFR2 or using the VEGFR3-selective ligand VEGF-C C156S, induced LEC proliferation and vessel dilation in vivo, it was not accompanied by the formation of new vessel sprouts and branches. While we did not examine sprouting at the cellular level (i.e., filopodia formation or migration of the sprouting front), formation of the superficial lymphatic capillary plexus arises by sprouting from pre-existing collecting vessels[21,23], and AAV9-mediated VEGF-C overexpression similarly induces robust abluminal sprouting. Notably, both sprouting-dependent processes were inhibited by *Vegfr2* deletion. Together, these findings show that VEGFR3 homodimer activation promotes proliferation without sprouting, whereas VEGFR2 homodimer activation alone induces neither response, suggesting that functional lymphangiogenic sprouting requires coordinated activation of both receptors.

How VEGFR2 contributes mechanistically to lymphangiogenic sprouting remains less clear. A plausible mechanism by which VEGFR2 may regulate VEGF-C-induced lymphangiogenesis is through the regulation of cell surface availability of VEGFR3. Angiopoietin/Tie signalling was recently shown to maintain cell surface levels of VEGFR3 through PI3Kα[30], which is known to promote endocytic recycling of receptors[37]. In a similar scenario, loss of VEGFR2/3-PI3Kα signalling would lead to a failure in VEGFR2/3 recycling and ultimately to the loss of VEGFR2/3 on the cell surface, thereby impairing the ability of LECs

to respond to VEGF-C. However, several of our findings argue against such a model as the primary explanation for the distinct lymphatic phenotypes observed in vivo. *Vegfr2* deletion did not affect the cell surface levels of VEGFR3 during development or adult homeostasis, indicating that VEGFR2 is not simply required to maintain VEGFR3 surface availability. Consequently, in the absence of *Vegfr2*, LECs retained the ability to proliferate in response to VEGF-C, whereas sprouting and migration were selectively impaired. Finally, we observed differential regulation of VEGF receptor abundance downstream of VEGF-C stimulation and PI3Kα signalling. VEGFR3 undergoes internalization upon VEGF-C stimulation and requires PI3Kα for its baseline maintenance at the plasma membrane, whereas steady-state VEGFR2 levels were unaffected by *Pik3ca* loss, yet VEGF-C induced a PI3Kα-dependent increase in VEGFR2 abundance.

Another possibility is that VEGFR2 directly modulates VEGFR3 signalling through VEGF-C-induced formation of VEGFR2/VEGFR3 heterodimers, which in vitro generate a distinct VEGFR3 phosphorylation pattern compared to VEGFR3 activation[13]. In vitro studies in primary LECs have further shown that VEGFR3 homodimers and VEGFR2/3 heterodimers activate distinct downstream signalling pathways[14] and different signalling kinetics[39]. For example, VEGFR2 knockdown selectively reduced AKT, but not ERK, activation upon VEGF-C stimulation[14], suggesting that PI3K-AKT is the primary downstream pathway for VEGFR2/3 heterodimers. However, different conclusions have been drawn from studies using function-blocking antibodies that interfere with ligand binding or receptor (hetero) dimerization[40]. In these experiments, both VEGFR2 and VEGFR3 were proposed to contribute to ERK activation, whereas AKT activation was primarily induced by VEGFR3[40]. The reasons for these discrepancies, and the potential in vivo role of a VEGFR2/3 heterodimer, remain unclear, likely because conclusions from in vitro studies of VEGFR2 signalling are limited in their relevance to the in vivo context as discussed above. Importantly, although these functional in vitro studies strongly imply receptor co-operation consistent with a heterodimer-based signalling model, neither they nor our current in vivo data provide direct biochemical proof of heterodimer signalling. PLA assays performed here and previously by others[41] in developing dermal and adult tracheal lymphatic vessels, respectively, demonstrate close spatial proximity between VEGFR2 and VEGFR3 in vivo, again consistent with, but not definitive proof of, heterodimerization. Nevertheless, our in vivo data showing distinct cellular responses of VEGFR3 alone, promoting LEC proliferation, versus the combined requirement of VEGFR2 and VEGFR3 for sprouting, together with in vivo PLA evidence, strongly support a model in which VEGF-C induces functional VEGFR2/3 heterodimers. In line with a major role of VEGFR2/3–PI3K–AKT signalling in lymphatic sprouting, activating mutations in *PIK3CA* encoding PI3Kα, which cause lymphatic malformation in humans[42,43], lead primarily to LEC migration and lymphatic vessel sprouting in mice[18,43]. Conversely, as shown in our current study, genetic deletion of *Pik3ca* completely abrogated VEGF-C-induced lymphatic hypersprouting.

Based on these findings, we propose a model in which VEGF-C regulates both receptor abundance and downstream signalling through coordinated activation of VEGFR3 homo- and VEGFR2/3 heterodimers. In this model, VEGF-C initially binds to its high-affinity receptor VEGFR3[44], triggering its internalization and LEC proliferation. This reduces cell surface levels of VEGFR3 homodimers while concomitantly increasing VEGFR2 abundance in a PI3Kα-dependent manner, thereby priming LECs for sprouting through enhanced VEGFR2/3 heterodimer formation. Such coordination may couple LEC proliferation with vessel sprouting, ensuring that sufficient cell numbers are available to support the formation of functional lymphatic vessels (Fig. 8). Furthermore, the proteolytic processing of VEGF-C, which can occur after VEGFR3 binding[45] and increases its affinity for VEGFR2[44], may reinforce the dynamic balance between receptor homo- and heterodimer formation, thus coordinating proliferation versus sprouting.

Interestingly, recent studies in retinal angiogenesis showed that blood ECs exhibit a bell-shaped response to mitogenic ERK signalling, whereby high levels of VEGF stimulation favour migration and sprouting while suppressing proliferation[46]. These findings highlight the importance of ligand concentration and signalling duration, with acute, transient stimulation promoting balanced vascular growth, whereas chronic, high-dose VEGF exposure induces aberrant endothelial behaviour. Accordingly, VEGF-C–mediated tuning of VEGFR3 and VEGFR2/3 signalling may function to temporally separate proliferative and sprouting responses to support productive lymphangiogenesis. However, in our experimental setting, VEGF-C is delivered at unphysiologically high levels via AAV-mediated overexpression, and within the timeframe analysed, we cannot strictly separate temporally distinct proliferative and sprouting phases; instead, these processes are likely to occur in parallel. Nonetheless, early time-point analyses indicated that VEGF-C can induce LEC proliferation before sprouting, whereas PI3Kα activation is associated with lymphatic sprouting.

In line with the concept that endothelial responses are highly sensitive to signalling magnitude and heterogeneity, and consistent with our previous work[9], inducible postnatal deletion of *Vegfr3* paradoxically led to aberrant patterning and hyperplasia of the dermal lymphatic vasculature of the ear, driven by residual LECs that escaped recombination. One contributing factor may be the local accumulation of VEGF-C, resulting from inefficient ligand uptake by the hypoplastic lymphatic vasculature caused by LEC-autonomous loss of *Vegfr3*, which was shown to control capillary density by driving side-branching[23]. In addition, our prior study showed that a major contributor to these pathological effects is non–cell-autonomous: *Vegfr3*-deficient LECs promote proliferation of neighbouring VEGFR3+ LECs through cell–cell contact–dependent suppression of Notch signalling. In contrast, *Vegfr2* deletion did not induce comparable non–cell-autonomous effects or abnormal behaviour of residual wild-type LECs, suggesting a unique role for VEGFR3 in Notch-mediated lateral signalling that coordinates LEC behaviour during vessel growth and patterning.

Taken together, our study uncovers a previously unknown critical function of VEGFR2 in VEGF-C–VEGFR3-mediated lymphatic vessel sprouting, both during development and in neo-lymphangiogenic growth in adult tissues. This finding has important therapeutic implications, suggesting that effective stimulation of functional lymphangiogenesis requires both VEGFR2 and VEGFR3.

## Methods

### Mouse lines and treatments

Heterozygous null *Vegfr2*[+/-] allele was generated by crossing *Vegfr2*[flox] (*Kdr*[tm1Wag]) mice[47] with mice carrying the *PGK-Cre* transgene[48]. *Vegfr2*[+/-] animals were further backcrossed to *Vegfr2*[flox] (*Kdr*[tm1Wag]) mice and mice carrying the *R26-mTmG* reporter[25] and the LEC-specific *Prox1-CreER*[T2] transgene[24]. Postnatal gene deletion was induced by three, five or ten consecutive topical administrations of 15 μl of tamoxifen (#T5648, Sigma), dissolved in acetone (10 mg ml⁻¹), to the abdomen (until P8) or ear skin. For high-fidelity gene deletion and cell tracking, *Vegfr2*[flox](*Kdr*[tm1Sato]) mice[49] were crossed with mice carrying the *iSuRe-Cre*[26] or *R26-iSuRe-HadCre*[11] and the *Prox1-CreER*[T2] transgene[24], or the *Vegfr1-CreER*[T2] transgene[18]. Mosaic postnatal gene deletion was induced by intragastric (i.g.) injection of 2 μl (50 μg) of 4-hydroxytamoxifen (4-OHT) (#H7904, Sigma), dissolved in absolute ethanol (25 mg ml⁻¹), at P1 and P2. Gene deletion in post-weaning aged mice was induced by oral gavage of 100 μl (1 mg) or 200 μl (2 mg) of tamoxifen, dissolved in peanut oil (10 mg ml⁻¹), at one, three or five consecutive days. Tracing of *Vegfr2* functional mosaics was performed by crossing *R26-iMb-Vegfr2* mice[27] to *Cdh5-CreER*[T2] mice[28], and tracing

was induced by intragastric (i.g.) injection of 2 µl (50 µg) of 4-OHT, dissolved in absolute ethanol (25 mg ml$^{-1}$), at P0 or P1. Genetic deletion of *Vegfr2* and/or *Vegfr3* was induced neonatally in *Vegfr2$^{flox}$;Vegfr3$^{flox}$;R26-iSuRe-HadCre;Prox1-CreER$^{T2}$* or in 3-week-old *Vegfr3$^{flox}$;R26-mTmG;Prox1-CreER$^{T2}$* mice[9] as described above. Embryonic deletion was induced by intraperitoneal injection of 100 µl (1 mg) of 4-OHT, dissolved in peanut oil (10 mg ml$^{-1}$), into pregnant dams. The day of the vaginal plug was considered E0.5. LEC-specific PI3Kα activation was induced in *R26-LSL-Pik3ca$^{H1047R}$;Vegfr3-CreER$^{T2}$* mice by topical application of 50 µg of 4-OHT, dissolved in acetone (10 mg ml$^{-1}$), onto the dorsal side of each ear as previously described[18]. For LEC-specific PI3Kα inactivation, *Pik3ca$^{flox}$* mice[50] were used in combination with the *Prox1-CreER$^{T2}$*. Ligand overexpression was achieved by intradermal injection of $2.5 \times 10^{10}$ viral particles of AAV9-VEGF-C[15], AAV9-VEGF-CΔNΔC[51], AAV9-VEGF-C156S[15] or $2.0 \times 10^9$ viral particles AAV9-VEGF-A$_{164}$[15] dissolved in 10 µl PBS, into the dorsal side of the ear of anaesthetized mice. Systemic inhibition of VEGF-C using VEGF-C trap was achieved via intraperitoneal injection of $1 \times 10^{11}$ viral particles AAV9-VEGFR3[1-4]-Ig dissolved in 100 µl PBS (adult) or $5 \times 10^{10}$ viral particles AAV9-VEGFR3[1-4]-Ig dissolved in 20 µl PBS (neonatal). All mice were analyzed on a C57BL/6 J Jax background, and both female and male mice were used. Animal experiments were approved by the Uppsala Animal Experiment Ethics Board (permit numbers 130/15, 06383/2020), or the Finnish Project Authorisation Board (Eläinkoelautakunta ELLA, permit numbers ESAVI/28504/2022 and ESAVI/15313/2025), and were done in accordance with national and EU regulations.

## VEGF ELISA
Dorsal ear skin of mice injected with AAV9-VEGF$_{164}$ was dissected for microscopic analysis whereas the ventral ear skin containing the intermediate cartilage layer was flash frozen in liquid nitrogen. Tissue was subsequently minced using mechanical dissociation on ice in RIPA buffer containing 1 × Halt Protease & Phosphatase inhibitor cocktail (#78440, ThermoFisher Scientific) using TissueRuptor® II (Qiagen). VEGF concentration was determined using the Quantikine ELISA Mouse VEGF kit (#MMV00, R&D systems) according to the manufacturer's instructions. VEGF overexpression was considered successful in mice with >200 pg mg$^{-1}$ total protein (range 206-2388 pg mg$^{-1}$, $n = 13$ ears from 7 mice). Absorbance was measured using the Agilent BioTek Synergy LX plate reader (Fisher Scientific).

## Flow cytometry
Ear skin was dissected and digested in 2 ml of digestion buffer containing 5 mg ml$^{-1}$ collagenase II (Sigma), 0.2 mg ml$^{-1}$ DNaseI (Roche) and 0.2% FBS (Gibco) in 1 × PBS for 10 min at 37 °C and 950 RPM shaking. Digestion was quenched by the addition of 10 µl of 0.5 M EDTA. Cell suspension was filtered through a nylon 50 µm CellTrics filter (Sysmex) and diluted further to a total volume of 5 ml with FACS buffer containing 0.5% FBS and 2 mM EDTA in PBS. Cells were subsequently pelleted by centrifugation and resuspended in 200 µl of fresh FACS buffer. Unspecific binding of antibodies to Fc receptors was prevented by blocking with CD16/32 antibodies (#14-0161-85, ThermoFisher). Live endothelial cells expressing tdTomato (Fig. 3e) were stained using Podoplanin-APC (#127410, Biolegend), CD31-Pe-Cy7 (#25-0311-82, ThermoFisher) and LYVE1-AF488 (#53-0443-82, ThermoFisher). Immune cells, red blood cells and dead cells were excluded by staining with CD45-eF450 (#48-0451-82, ThermoFisher), CD11b-eF450 (#48-0112-82, ThermoFisher) and Ter119-eF450 (#48-5921-8, ThermoFisher) antibodies, and SYTOX Blue dead stain (#S11348, Life Technology), respectively. For analysis of KI67+ proliferating cells (Fig. 5f,g), unspecific binding of antibodies to Fc receptors was prevented by blocking with CD16/32 antibodies CD16/32 antibodies (#14-0161-85, ThermoFisher) followed by staining using CD45-eF450, CD11b-eF450 (to exclude immune cells), CD31-PerCP-eF710 (#46-0311-80, ThermoFisher), PDPN-PE-Cy7 (#25-5381-82, ThermoFisher),

followed by staining of live and dead cells using blue LIVE/DEAD® fixable dead cell stain kit (#L23105, ThermoFisher) and subsequent fixation and permeabilization using FOXP3 Fix/Perm Buffer Set (#00-5523-00, ThermoFisher). Finally, samples were stained for intracellular KI67 using KI67-eF660 (#50-5698-80, ThermoFisher). All flow cytometry experiments were analyzed on BD LSRFortessaTM Cell Analyzer (BD Biosciences). Flow cytometry analysis of proliferation was performed as described previously[18].

## Whole-mount immunofluorescence
Dorsal ear skin was separated from the underlying cartilage layer and fixed in 4% PFA dissolved in PBS for 1 h or 2 h at room temperature (RT). Embryonic back skin was dissected from the underlying muscle layer, pinned down in Syllguard plates, and fixed in 4% PFA for 2 h at RT. Tissue was washed in PBST (PBS + 0.3% TritonX-100) for 15 min and blocked for 2 h at RT in PBST containing 5% BSA and 1% FBS or PBST plus 3% non-fat milk. Samples were subsequently stained using the following antibodies: goat anti-mouse VEGFR3 (1:100) (#AF743, R&D Systems), goat anti-mouse VEGFR2 (1:100) (#AF644, R&D Systems), rat anti-mouse LYVE1 (1:200) (#MAB2125, R&D systems), rabbit anti-mouse LYVE1 (1:500) (#103-PA50AG, Reliatech), rat anti-mouse PECAM1 (1:200) (#553370, BD Pharmingen), armenian hamster anti-mouse PECAM1 (1:200) (#MA3105, ThermoFisher), chicken anti-GFP (1:500) (#ab13970, Abcam), rabbit anti-DsRed (1:500) (#632496, Takara), syrian hamster anti-mouse PDPN (1:200) (#8.1.1, DSHB), rabbit-anti-human PROX1 (1:200) (#102-PA32AG, Reliatech), KI67 eFluor 660 (1:300) (#50-5698, eBioscience), alpha-Tubulin (1:200) (#ab52866, Abcam), Phalloidin-Atto 647 N (#65906, Sigma), diluted in PBST containing 3% BSA and 1% FBS or PBST plus 3% non-fat milk, overnight at 4 °C. Samples were washed for 2 h at RT in PBST with several buffer changes and stained with donkey anti-goat Alexa Fluor plus 647 (#A32849, ThermoFisher), donkey anti-rabbit Cy3 (#711-165-152, Jackson ImmunoResearch), donkey anti-rat Alexa Fluor 488 (#712-545-150, Jackson immunoResearch), donkey anti-chicken Alexa Fluor 488 (#703-545-155, Jackson ImmunoResearch), goat anti-armenian hamster Alexa Fluor 594 (#127-585-160, Jackson ImmunoResearch), donkey anti-goat IgG-Cy5 (#705-175-003, Jackson ImmunoResearch), donkey anti-goat Alexa Fluor 594 (#A32758, ThermoFisher), donkey anti-rabbit Alexa Fluor 488 (#A32790, ThermoFisher), donkey anti-rabbit Alexa Fluor 647 (#A32795, ThermoFisher), donkey anti-rat Alexa Fluor 488 (#A48269, ThermoFisher), donkey anti-goat Alexa Fluor 555 (#A32816, ThermoFisher), donkey anti-rabbit Alexa Fluor 594 (#A32754, ThermoFisher), donkey anti-rat Alexa Fluor 594 (#A21209, ThermoFisher), or donkey anti-rabbit Alexa Fluor 647 (#A32794, ThermoFisher) secondary antibodies, all diluted 1:500 or 1:300 in 3% BSA plus 1% FBS in PBST or PBST plus 3% non-fat milk for 2 h at RT. After final washes at RT, the samples were mounted in Mowiol. For assessing cell surface levels of protein, PBST was substituted with PBS to preserve membrane integrity.

## Punch biopsy model of lymphatic regeneration
A 2-mm hole was made in the centre and close to the ear tip in both ears of each mouse by using a metal ear puncher (#AT7000, AgnThos). The ear skin was then collected and fixed in 4% PFA for 2 h at RT, one week after the hole punching, followed by whole mount immunofluorescence analysis.

## Whole-mount Proximity ligation assay (PLA)
Embryos were fixed for 2 h in 4% PFA dissolved in TBS at RT. The back skin was dissected from the underlying muscle layer, pinned down on Syllguard plates, and fixed in 4% PFA for an additional 1 h at RT. Tissue was permeabilized in TBS containing 0.3% TritonX-100 (TBSTx) for 1 h at RT. PLA assay was performed using NaveniFlex GR PLA kit (Navinci) according to the kit's protocol with the following adaptations: all sample processing, except for washing, was done in 96-well plates, and

washing times and signal development during oligo annealing was increased. Goat anti-mouse VEGFR3 (1:100) (#AF743, R&D Systems), goat anti-mouse VEGFR2 (1:100) (#AF644, R&D Systems), rabbit anti-VEGFR2 (1:100) (#2479, Cell Signalling), and rabbit anti-phospho-Tyrosine (P-Tyr-1000) MultiMab (1:100) (#8954, Cell Signalling) primary antibodies were used for generating PLA signals. The samples were post-fixed in 1% PFA in TBS for 15 min after which primary antibodies used for counterstaining were added and incubated overnight at 4 °C, and the whole mount immunofluorescence protocol described above was followed for detection and mounting.

## Microscopy and image analysis

All imaging was performed at RT using a Leica SP8 or Leica Stellaris 5 confocal microscope using a 10 × /0.45 C-Apochromat (HC PL APO CS2), 20 × /0.75 (HC PL APO CS2), 25 × /0.95 (HC FLUOTAR L VISIR), or a 63 × /1.20 (HC PL APO) objective and the LasX software. Figure 2a was imaged using a Leica SP8 DIVE multiphoton microscope equipped with a Ti:Sapphire multiphoton laser emitting a 680–1300 nm tunable and 1045 nm fixed laser line, using a HC IRAPO 25 × /1.0 objective. Images were processed using Fiji/Image J software (NIH). All images represent maximum intensity projection of z-stacks. For comparison of staining intensities, the same parameters were used for image acquisition and processing. For quantification of vessel parameters, tile scan images of P21 or 5-week ear skin were used. Vessel ends were counted manually using Photoshop CS6 (Fig. 2g) or Fiji/Image J (NIH) (Fig. 4i) and normalized to tissue area. The central area was defined as excluding vessel tips terminating at the tissue edge. Branch points and tip occupancy in Fig. 2e,f,h were also counted manually. Vessel area and branch points in Figs. 3f, 4g, h, 5d were determined automatically using the published script[52] for vessel analysis utilizing MATLAB (MathWorks, version R2022). Collecting vessel diameter in Fig. 1d was quantified manually using Fiji/ImageJ using the line tool. Average diameter was quantified by measuring the diameter of each individual lymphangion segment (vessel segment between two valves) on collecting vessels. The number of capillary vessel ends showing different morphology in Fig. 5h,i was counted manually and separately, and normalized to the total vessel end number. The number of blunt vessel ends (Fig. 5h,i) or sprouts (Supplementary Fig.4b) containing KI67+ cells was counted manually and normalized to the total number of blunt-ended vessels or the total number of sprouts. For the quantifications of the punch assay in Fig. 6b,c, the spiky vessel ends were counted manually and normalized to the wound region (200 μm from the punched edge) of the ear, and lymphatic vessel area was measured using Fiji/Image J (NIH) and normalized to the wound region (100 μm from the punched edge) of the ear. Quantification of VEGFR2 staining intensity in Fig. 7f and Supplementary Fig. 7 was done by measuring corrected total cell fluorescence (CTCF) = integrated density − (area of selected cell × mean fluorescence of background readings) using Fiji/ImageJ (NIH).

## Statistical analysis

Data was analyzed using Prism (version 9 and 10) for MacOS software (version 9.0.0 and 10.6.1). Comparison between two groups was done using the Mann-Whitney U test. For comparison of more than two groups, one-way ANOVA followed by Tukey's multiple comparisons test was used. Categorial variables were compared using Fisher's exact test. $p$ values < 0.05 were considered significant and depicted as $*p < 0.05$, $**p < 0.01$, $***p < 0.001$ and $****p < 0.0001$.

## Reporting summary

Further information on research design is available in the Nature Portfolio Reporting Summary linked to this article.

## Data availability

All source data supporting the quantitative findings of this study are provided as a Source Data file. All other data supporting the findings are available within the paper and its supplementary information files. Source data are provided with this paper.

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

## Acknowledgements

We thank Ralf Adams (Max Planck Institute for Molecular Biomedicine, Münster) for the *Cdh5-CreER^{T2}* mice, Bart Vanhaesebroeck (UCL Cancer Institute, University College London, London) and Mariona Graupera (Josep Carreras Leukaemia Research Institute, Barcelona) for the *Pik3ca^{flox}* mice, and Sofie Lunell Segerqvist and Charlotte Rorsman (Uppsala University) and Maria Arraño de Kivikko and Mari Jokinen (Wihuri Research Institute, Helsinki) for technical assistance. Kari Alitalo, Tanja Laakkonen and Andrey Anisimov (University of Helsinki) are acknowledged for AAV vectors, produced in the AAV Core Unit supported by the University of Helsinki (HiLIFE and Research Programs Unit, Faculty of Medicine) and Biocenter Finland. Kari Alitalo, Pipsa Saharinen, Kari Vaahtomeri and Ingvar Ferby are acknowledged for helpful discussions and critical comments on the manuscript. This work was supported by grants from Knut and Alice Wallenberg Foundation (2018.0218 and 2020.0057), the Swedish Research Council (2020-02692), Göran Gustafsson foundation, the Swedish Cancer Society (19 0220 Pj, 22 2025 Pj), Sigrid Juselius Foundation (8193), the Research Council of Finland (371210, 374177) and Jenny and Antti Wihuri Foundation (all to TM), as well as by the European Union's Horizon 2020 research and innovation programme under the Marie Skłodowska-Curie grant agreement No 814316 (to HS, TM) and the European Research Council Consolidator Grant AngioUnrestUHD (to RB).

## Author contributions

H.S. conceived and designed the study, performed experiments, analyzed and interpreted data, and wrote the manuscript. Y.Z. performed experiments on initial neonatal *Vegfr2* deletion, punch hole wound healing, early time-point analysis of vessel morphology and proliferation and receptor abundance, as well as analyzed and interpreted data. H.O. performed experiments and analyzed data on flow cytometric analyses. M.L. performed experiments on *R26-iMb-Vegfr2* mice. R.B. provided mouse lines and related expertise. T.M. conceived and designed the study, analyzed and interpreted data, supervised the project and wrote the manuscript. All authors discussed the results and commented on the manuscript.

## Competing interests

The authors declare no competing interests.
