## [Transparent Peer Review file · Nature Communications]

VEGFR2 is required for VEGF-C–VEGFR3–PI3K α -mediated sprouting lymphangiogenesis

Corresponding Author: Professor Taija Makinen

Version 0:

Reviewer comments:

Reviewer #

(Remarks to the Author)

VEGF-C/VEGFR3 signaling is a key regulator of lymphangiogenesis in both health and disease. However, the role of VEGFR2—the primary receptor mediating VEGF-A-driven angiogenesis—in lymphatic endothelial cells (LECs) remains poorly understood. To investigate this unresolved question, the authors utilized several mouse models with VEGFR2 and/or VEGFR3 gene knockouts, along with advanced tools for tracing Cre-targeted cells. The main finding of this study is that the loss of VEGFR2 reduces the competitive potential of individual LECs during the formation of lymphatic capillaries, but not during the development of collecting vessels. Notably, the effects of VEGF-A overexpression on lymphatic vessels appear to be secondary to blood vascular leakage and the recruitment of VEGF-C–producing macrophages.

Overall, this paper clearly enhances our understanding of the mechanisms by which VEGF receptors coordinately regulate the development of lymphatic vessels. The reviewer has only a few minor comments:

. (Title & abstract) “Sprouting” refers to a biological process in which cells extend outgrowths or branches to form new connections or structures. However, the data presented in this paper only examines the formation of superficial lymphatic capillaries, not the sprouting process itself. In my opinion, the title and abstract are somewhat overstated and should be revised accordingly.

2. (line 57-59) Dellinger et al. (PLoS One, 2013) demonstrated that Lyve1-Cre-mediated deletion of Vegfr2 leads to a reduced number of lymphatic vessels. This study highlights the significant role of Vegfr2 in lymphatic vessel development and should therefore be discussed in the context of the current findings.

3. (line 184-186) The statement "Although collecting vessels formed by Tom+VEGFR2- LECs ... downregulated the lymphatic capillary marker LYVE1 (Fig. 3c)" is not clearly supported by Fig. 3c. Could the authors provide a high magnification view of LYVE1 channel associated with Fig. 3d? This would help confirm the presence of lymphatic vessels, as indicated by the dashed lines in the lower right panel.

4. (Fig. 6h) As mentioned in the discussion, there is no evidence provided in this paper showing the functional relevance of the R2/R3 heterodimer. Therefore, the schema should be modified accordingly.

Reviewer #2

(Remarks to the Author)

Using a multifaceted approach that includes inducible genetically modified mice and AAV-mediated delivery of specific VEGFR ligands, the authors convincingly demonstrate the critical role of VEGFR2 in VEGF-C–VEGFR3-driven lymphatic vessel sprouting during both developmental and neo-lymphangiogenic vascular growth in the ear skin. They show that activating VEGFR2 alone is not enough to induce lymphangiogenesis. In neonates, VEGFR2, along with VEGFR3, is essential for lymphatic capillary sprouting and LEC survival. In adults, loss of VEGFR2 prevents VEGF-C-induced lymphatic sprouting without affecting LEC proliferation, while genetic deletion of VEGFR3 or its downstream effector PI3K α completely halts lymphangiogenesis. They utilize advanced techniques to demonstrate that VEGFR2 is activated and physically associates with VEGFR3, with the surface presence of both receptors depending on PI3K α . Based on these findings, the

authors claim that VEGFR2 mediates a balance between VEGF-C-induced LEC proliferation and lymphatic sprouting, supporting functional lymphangiogenesis through a shared signaling pathway. Consequently, effective therapeutic strategies should target both VEGF-C receptors simultaneously.

Overall, this is a solid study demonstrating the critical role of VEGFR2 in VEGF-C–VEGFR3-driven lymphangiogenesis during both developmental and neo-lymphangiogenic vascular growth. This paper is well-written, well-designed, well-executed, and well-analyzed. My only major comment is the following:

Major Comment

. At the end of the Summary and Discussion, the authors addressed "the effective therapeutic strategies for lymphangiogenesis," but there is no supporting data for this claim in the study. Therefore, this reviewer recommends that they also include some evidence of regenerative lymphangiogenesis following ear skin injury caused by a punch hole biopsy.

Minor Comments

- . Two Discussions in p5 and p14.
2. Please replace "Box line" in Fig. 6c with a color that stands out.

Reviewer #3

(Remarks to the Author)

Vascular endothelial growth factor receptor 2 (VEGFR2) is a receptor expressed by LECs. However, its function in LECs has not been widely investigated. A previous study using Lyve1-Cre mice revealed that loss of Vegfr2 in Lyve1-lineage cells impaired the growth, but not the maturation, of lymphatic vessels. Here, the authors perform elegant genetic experiments to further identify the function of VEGFR2 in lymphatic vessels. They demonstrate that LECs lacking Vegfr2 do not contribute to lymphatic sprouts as readily as wild-type LECs. Furthermore, they show that continuous administration of tamoxifen in Prox1-CreERT2;Vegfr2^{f/f} mice significantly impairs the branching of lymphatic vessels. The experiments are well-designed and provide valuable mechanistic insight into the function of VEGFR2 in lymphatics. I believe that the results presented in this paper will be well received by the general readership of Nature Communications.

Major Comments

. In numerous sections of the manuscript, the authors state that Vegfr2 has been considered dispensable for lymphangiogenesis. However, a previous study showed that conditional knockout of Vegfr2 in LECs impaired the development of lymphatic vessels in the trachea (PMID: 35050301). Additionally, a separate report showed that conditional knockout of Vegfr2, using a Lyve1-driven constitutive Cre, impaired lymphatic branching but not the development of collecting vessels or valves (PMID: 24023956). These prior studies should be included in the introduction of the manuscript, especially given the authors' observation that loss of Vegfr2 in lymphatic endothelial cells impairs lymphatic branching, but not the development of collecting vessels or valves, which aligns with a previous report (PMID: 24023956).

2. The diameter of lymphatic vessels in Figure 1E should be quantified and compared to appropriate control groups (e.g., Vegfr2^{f/f};Vegfr1-CreER w/o AAV-VEGF-A, Vegfr2^{f/f};Vegfr1-Cre with AAV-VEGF-A, Control with VEGF-A) to demonstrate that sole activation of Vegfr2 on LECs does not impact lymphatic diameters.

3. The experiment in Figure 3G is elegant and further suggests that LECs lacking Vegfr2 activity do not contribute to lymphatic capillaries/sprouts as readily as wildtype LECs. The major limitation of the experiment is that the authors only analyzed -2 mice. Additional mice (at least three) should be examined to increase the robustness of the observation.

4. Figure 6D is confusing as displayed. It is unclear which Cre lines were used and when tamoxifen was given. This information should be added to the manuscript. Furthermore, it appears as though the authors didn't examine VEGFR2 localization in Pik3caf/f mice in the absence of AAV-VEGF-C. This result should be shown to determine whether the localization of VEGFR2 is impacted in Pik3caf/f mice or if differences are only observed when high levels of VEGF-C are present. The authors should also indicate in the figure legends the number of mice they analyzed.

. The authors' data suggest that loss of Pik3ca impairs the localization of VEGFR2 and VEGFR3. This observation is primarily demonstrated by whole-mount immunofluorescence staining. To strengthen this observation, the authors should examine the localization of VEGFR2 and VEGFR3 by ICC or FACS in cultured LECs after knocking down of Pik3ca or treating the cells with a PI3K inhibitor, such as alpelisib.

. It is unclear how many mice were analyzed in Figures 6B and 6C. This information should be added to the figure legend. The authors rely on the comparison of permeabilized to unpermeabilized samples to compare total to surface levels of VEGFR2 and VEGFR3. The authors should stain permeabilized and unpermeabilized samples for a cytoplasmic or nuclear marker (e.g., Prox1) to demonstrate that intracellular proteins are not detected in unpermeabilized samples. This experiment is required to support the validity of the technique. The result from the experiment could be added as a supplemental figure.

Minor Comments

- . The authors should include in the figure legend for Supplemental Figure how many mice they analyzed per group.
2. It is unclear in Figure 2, panels F through H, whether the graphs should be labeled “flox/-” or “flox/flox”.

Version :

Reviewer comments:

Reviewer #

(Remarks to the Author)

I have carefully reviewed the point-by-point responses in the revised manuscript. The authors have adequately addressed my previous concerns and strengthened the data. The manuscript is now acceptable for publication.

Reviewer #2

(Remarks to the Author)

The authors adequately addressed the comments. Congratulations on this excellent study!

Reviewer #3

(Remarks to the Author)

The authors have addressed my comments.

NCOMMS-25-41000-T Response to the reviewers' comments

We thank the Reviewers for their insightful and constructive comments. In response, we have revised the manuscript and incorporated new experimental data. The figure panels containing new data are highlighted in **bold** in our point-by-point responses below.

Reviewer #1 (Remarks to the Author):

VEGF-C/VEGFR3 signaling is a key regulator of lymphangiogenesis in both health and disease. However, the role of VEGFR2—the primary receptor mediating VEGF-A-driven angiogenesis—in lymphatic endothelial cells (LECs) remains poorly understood. To investigate this unresolved question, the authors utilized several mouse models with VEGFR2 and/or VEGFR3 gene knockouts, along with advanced tools for tracing Cre-targeted cells. The main finding of this study is that the loss of VEGFR2 reduces the competitive potential of individual LECs during the formation of lymphatic capillaries, but not during the development of collecting vessels. Notably, the effects of VEGF-A overexpression on lymphatic vessels appear to be secondary to blood vascular leakage and the recruitment of VEGF-C-producing macrophages.

Overall, this paper clearly enhances our understanding of the mechanisms by which VEGF receptors coordinately regulate the development of lymphatic vessels. The reviewer has only a few minor comments:

We thank the reviewer for the positive feedback and for recognizing the contribution of our study.

1. (Title & abstract) “Sprouting” refers to a biological process in which cells extend outgrowths or branches to form new connections or structures. However, the data presented in this paper only examines the formation of superficial lymphatic capillaries, not the sprouting process itself. In my opinion, the title and abstract are somewhat overstated and should be revised accordingly.

The formation of the superficial lymphatic capillary plexus is a direct result of sprouting from pre-existing (pre-)collecting vessels^{1,2}. Similarly, the lymphatic response to AAV9-mediated VEGF-C overexpression is characterized by robust abluminal sprouting from pre-existing vessels, which we now show more clearly with early time point analysis (**Fig. 5h,i, Supplementary Fig. 4**). Thus, although we do not directly assess sprouting at the cellular level (i.e. filopodia formation or migration at the sprouting front), our results show that these two sprouting-dependent processes are inhibited by *Vegfr2* deletion. We now explicitly state this in the discussion (lines 446-450).

2. (line 57-59) Dellinger et al. (PLoS One, 2013) demonstrated that *Lyve1-Cre*-mediated deletion of *Vegfr2* leads to a reduced number of lymphatic vessels. This study highlights the significant role of *Vegfr2* in lymphatic vessel development and should therefore be discussed in the context of the current findings.

We agree and have now included discussion of the findings by Dellinger et al (lines 419-421). In this study, a hypoplastic lymphatic capillary plexus was indeed observed in *Vegfr2*^{lox/lox}; *Lyve1-Cre* mice, while valve formation and collecting vessel maturation

appeared largely normal, supporting an important role for VEGFR2 in lymphatic vessel development. However, *Lyve1-Cre* is not LEC-specific but also recombines in several blood vascular beds, including the yolk sac, liver and lung³. Loss of VEGFR2 in these tissues in *Vegfr2^{lox/lox}; Lyve1-Cre* was shown by Dellinger et al to severely impair vascular development, likely contributing to the reduced survival reported for these mutant embryos, with most embryos dying after embryonic day 14.5. The small fraction of surviving “escaper” animals (approximately 10% of the expected genotype) complicates interpretation of a strictly cell-autonomous role for VEGFR2 in LECs, as confounding effects arising from blood vascular defects cannot be excluded. These limitations of the study are now also discussed (lines 421-427).

3. (line 184-186) The statement "Although collecting vessels formed by Tom+VEGFR2-LECs ... downregulated the lymphatic capillary marker LYVE1 (Fig. 3c)" is not clearly supported by Fig. 3c. Could the authors provide a high magnification view of LYVE1 channel associated with Fig. 3d? This would help confirm the presence of lymphatic vessels, as indicated by the dashed lines in the lower right panel.

A higher magnification view of the LYVE1 channel of the images presented in **Fig. 3d** has now been provided.

4. (Fig. 6h) As mentioned in the discussion, there is no evidence provided in this paper showing the functional relevance of the R2/R3 heterodimer. Therefore, the schema should be modified accordingly.

We agree with the reviewer that our study does not provide direct biochemical evidence for VEGFR2/VEGFR3 heterodimer formation. In response, we have clarified this limitation in the discussion (lines 484–489) and in the **Fig. 8** legend, and revised the schematic model (now **Fig. 8**) to include a question mark between VEGFR2 and VEGFR3, reflecting the remaining uncertainty regarding heterodimer formation.

Importantly, this limitation also applies to previous *in vitro* studies, which have relied on receptor-specific ligands, receptor knock-down or blocking antibody approaches^{4,5}. In addition, PLA assays performed by us and previously by others⁶ in developing dermal and adult tracheal lymphatic vessels demonstrate close spatial proximity between VEGFR2 and VEGFR3 *in vivo*, but do not provide definitive proof of heterodimer formation or signalling. Nevertheless, our *in vivo* data demonstrate distinct signaling outputs: VEGFR3 alone promotes LEC proliferation, whereas VEGFR2 together with VEGFR3 is required for sprouting, consistent with receptor cooperation in a heterodimer-based signaling model.

Our new early time-point analysis further supports this distinction. AAV-VEGF-C induced rapid sprouting accompanied by robust LEC proliferation, detectable both at new sprouts and in blunt-ended capillaries with ‘quiescent’ morphology (**Fig. 5h**, **Supplementary Fig. 4a,b**). In contrast, LEC-specific activation of PI3K α , which *in vitro* studies have suggested requires VEGFR2 (ref⁵), promoted sprouting with proliferating LECs largely confined to the sprouts (**Fig. 5i**, **Supplementary Fig. 4a**).

Reviewer #2 (Remarks to the Author):

Using a multifaceted approach that includes inducible genetically modified mice and AAV-mediated delivery of specific VEGFR ligands, the authors convincingly demonstrate the critical role of VEGFR2 in VEGF-C–VEGFR3-driven lymphatic vessel sprouting during both developmental and neo-lymphangiogenic vascular growth in the ear skin. They show that activating VEGFR2 alone is not enough to induce lymphangiogenesis. In neonates, VEGFR2, along with VEGFR3, is essential for lymphatic capillary sprouting and LEC survival. In adults, loss of VEGFR2 prevents VEGF-C-induced lymphatic sprouting without affecting LEC proliferation, while genetic deletion of VEGFR3 or its downstream effector PI3K α completely halts lymphangiogenesis. They utilize advanced techniques to demonstrate that VEGFR2 is activated and physically associates with VEGFR3, with the surface presence of both receptors depending on PI3K α . Based on these findings, the authors claim that VEGFR2 mediates a balance between VEGF-C-induced LEC proliferation and lymphatic sprouting, supporting functional lymphangiogenesis through a shared signaling pathway. Consequently, effective therapeutic strategies should target both VEGF-C receptors simultaneously.

Overall, this is a solid study demonstrating the critical role of VEGFR2 in VEGF-C–VEGFR3-driven lymphangiogenesis during both developmental and neo-lymphangiogenic vascular growth. This paper is well-written, well-designed, well-executed, and well-analyzed. My only major comment is the following:

We thank the reviewer for the positive and thoughtful evaluation of our study.

Major Comment

1. At the end of the Summary and Discussion, the authors addressed “the effective therapeutic strategies for lymphangiogenesis,” but there is no supporting data for this claim in the study. Therefore, this reviewer recommends that they also include some evidence of regenerative lymphangiogenesis following ear skin injury caused by a punch hole biopsy.

We thank the reviewer for this helpful suggestion. In response, we have performed an additional experiment assessing regenerative lymphangiogenesis following ear skin punch biopsy in mice with LEC-specific deletion of *Vegfr2*, *Vegfr3*, or both. These new data are now presented in **Fig. 6** and described in lines 325-339. Briefly, we observed a significant reduction in lymphatic vessel growth at the wound edge in *Vegfr2*-deficient mice, while regenerative lymphangiogenesis was completely abolished in the absence of *Vegfr3*. Together, these results provide experimental support for the statements in the discussion regarding the role of VEGFR2 in regenerative lymphangiogenesis.

Minor Comments

1. Two Discussions in p5 and p14.

This has been corrected.

2. Please replace “Box line” in Fig. 6c with a color that stands out.

Box line color has been changed to red (now Fig. 7c).

Reviewer #3 (Remarks to the Author):

Vascular endothelial growth factor receptor 2 (VEGFR2) is a receptor expressed by LECs. However, its function in LECs has not been widely investigated. A previous study using *Lyve1-Cre* mice revealed that loss of *Vegfr2* in *Lyve1*-lineage cells impaired the growth, but not the maturation, of lymphatic vessels. Here, the authors perform elegant genetic experiments to further identify the function of VEGFR2 in lymphatic vessels. They demonstrate that LECs lacking *Vegfr2* do not contribute to lymphatic sprouts as readily as wild-type LECs. Furthermore, they show that continuous administration of tamoxifen in *Prox1-CreERT2;Vegfr2^{f/f}* mice significantly impairs the branching of lymphatic vessels. The experiments are well-designed and provide valuable mechanistic insight into the function of VEGFR2 in lymphatics. I believe that the results presented in this paper will be well received by the general readership of Nature Communications.

We thank the reviewer for this positive and encouraging assessment of our work.

Major Comments

1. In numerous sections of the manuscript, the authors state that *Vegfr2* has been considered dispensable for lymphangiogenesis. However, a previous study showed that conditional knockout of *Vegfr2* in LECs impaired the development of lymphatic vessels in the trachea (PMID: 35050301). Additionally, a separate report showed that conditional knockout of *Vegfr2*, using a *Lyve1*-driven constitutive Cre, impaired lymphatic branching but not the development of collecting vessels or valves (PMID: 24023956). These prior studies should be included in the introduction of the manuscript, especially given the authors' observation that loss of *Vegfr2* in lymphatic endothelial cells impairs lymphatic branching, but not the development of collecting vessels or valves, which aligns with a previous report (PMID: 24023956).

We thank the reviewer for highlighting these important studies. We have now revised the text to expand the discussion on the findings by Dellinger et al., 2013³ (PMID: 24023956) and Karaman et al., 2022⁷ (PMID: 35050301) (lines 419-429). Briefly (see also answer to R1 #2), we refer to the hypoplastic lymphatic vessel network reported in *Vegfr2^{lox/lox}; Lyve1-Cre* mice, while highlighting limitations in interpreting a strictly cell-autonomous role for VEGFR2 in LECs in this model, given that *Lyve1-Cre* drives recombination in multiple blood vascular beds that are affected by *Vegfr2* deletion. We also discuss the study by Karaman et al., which used pan-endothelial *Cdh5-CreER^{T2}* to delete *Vegfr2* in early postnatal and adult vasculature, however, noting that concomitant blood vascular defects preclude exclusion of secondary effects on lymphatics, including in the trachea.

2. The diameter of lymphatic vessels in Figure 1E should be quantified and compared to appropriate control groups (e.g., *Vegfr2^{f/f};Vegfr1-CreER* w/o AAV-VEGF-A,

Vegfr2f/f;Vegfr1-Cre with AAV-VEGF-A, Control with VEGF-A) to demonstrate that sole activation of Vegfr2 on LECs does not impact lymphatic diameters.

We agree with the reviewer and have now quantified lymphatic vessel diameters. Measurements from wild-type mice treated with AAV-VEGF-A, as well as *Vegfr2^{flox/flox};Vegfr1-CreER^{T2}* with and without AAV-VEGF-A have been included in revised **Fig. 1d**.

3. The experiment in Figure 3G is elegant and further suggests that LECs lacking Vegfr2 activity do not contribute to lymphatic capillaries/sprouts as readily as wildtype LECs. The major limitation of the experiment is that the authors only analyzed 1-2 mice. Additional mice (at least three) should be examined to increase the robustness of the observation.

We have now included additional mice to strength these results, with in total n=4 mice analyzed.

4. Figure 6D is confusing as displayed. It is unclear which Cre lines were used and when tamoxifen was given. This information should be added to the manuscript. Furthermore, it appears as though the authors didn't examine VEGFR2 localization in *Pik3caf/f* mice in the absence of AAV-VEGF-C. This result should be shown to determine whether the localization of VEGFR2 is impacted in *Pik3caf/f* mice or if differences are only observed when high levels of VEGF-C are present. The authors should also indicate in the figure legends the number of mice they analyzed.

We apologize for the lack of clarity. The figure and the legend have now been revised to clearly indicate the Cre line used, the timing of tamoxifen administration, and the genotype and experimental treatment of each sample (**Fig. 7d,e**). The experiment was repeated to ensure consistent comparisons between genotypes and treatments, and data was quantified (**Fig. 7f, Supplementary Fig. 7**). We have now also included data showing VEGFR2 localization in *Pik3ca^{flox/flox};Prox1-CreER^{T2}* mice in the absence of AAV-VEGF-C. Notably, baseline VEGFR2 levels were not reduced upon *Pik3ca* deletion, in contrast to VEGFR3, which showed reduced levels, as previously reported⁸. However, we found that the VEGF-C-induced increase in VEGFR2 was completely abolished in the absence of *Pik3ca*. The figure legend has been updated to indicate the number of mice analyzed for each condition.

5. The authors' data suggest that loss of *Pik3ca* impairs the localization of VEGFR2 and VEGFR3. This observation is primarily demonstrated by whole-mount immunofluorescence staining. To strengthen this observation, the authors should examine the localization of VEGFR2 and VEGFR3 by ICC or FACS in cultured LECs after knocking down of *Pik3ca* or treating the cells with a PI3K inhibitor, such as alpelisib.

We thank the reviewer for the suggestion. We carefully considered complementing our *in vivo* findings with experiments in cultured LECs; however, we believe that such approaches would not reliably recapitulate the *in vivo* context relevant to this study. As discussed in the manuscript, *in vitro* assays have previously yielded conflicting or misleading results in LECs, for example regarding the effects of VEGF or sole VEGFR2 activation on LEC behaviour. In addition, regulation of VEGF-C signalling *in vivo* relies

on complex proteolytic processing and spatial availability that are difficult to model in culture, as only fully processed VEGF-C is commercially available.

Instead, we opted to strengthen the *in vivo* data further. Importantly, our *in vivo* whole-mount immunofluorescence data, now more clearly presented and quantitatively analyzed (**Fig. 7d-f**), demonstrate the effects of *Pik3ca* loss on VEGFR2/3 levels and localization, including new data showing the dependence of VEGF-C-induced VEGFR2 upregulation on PI3K signalling.

6. It is unclear how many mice were analyzed in Figures 6B and 6C. This information should be added to the figure legend.

The figure legend has been updated to indicate the number of mice analyzed for each condition.

The authors rely on the comparison of permeabilized to unpermeabilized samples to compare total to surface levels of VEGFR2 and VEGFR3. The authors should stain permeabilized and unpermeabilized samples for a cytoplasmic or nuclear marker (e.g., Prox1) to demonstrate that intracellular proteins are not detected in unpermeabilized samples. This experiment is required to support the validity of the technique. The result from the experiment could be added as a supplemental figure.

We thank the reviewer for suggesting this important control. We have now performed, under the same conditions, staining for tubulin and phalloidin and present the results showing negligible staining under unpermeabilized conditions in **Supplementary Fig. 6a**. We also stained for PROX1 and observed that, while the signal was generally absent in the majority of LECs in unpermeabilized tissue, a subset of cells showed a weak signal, likely due to partial disruption of membrane integrity during tissue processing and fixation. VEGFR2 and VEGFR3 exhibit clearly distinct patterns between the conditions, now more clearly shown in **Fig. 7d,e**, and consistent with a prior study assessing VEGFR3 distribution *in vivo*⁸, further supporting the validity of the approach.

Minor Comments

1. The authors should include in the figure legend for Supplemental Figure 1 how many mice they analyzed per group.

The number of mice analyzed per group has now been added in the figure legend for Supp. Fig. 1; number of mice analyzed per genotype: n=6 (Cre wt ctrl), n=4 (flox/+), n=5 (flox/flox).

2. It is unclear in Figure 2, panels F through H, whether the graphs should be labeled “flox/-” or “flox/flox”.

We thank the reviewer for pointing this out. The graphs in Fig. 2f–h were incorrectly labeled as flox/flox and should have been flox/-. This has now been corrected.

References

1. Mäkinen, T. *et al.* PDZ interaction site in ephrinB2 is required for the remodeling of lymphatic vasculature. *Genes Dev.* **19**, 397–410 (2005).
2. Uçar, M. C. *et al.* Self-organized and directed branching results in optimal coverage in developing dermal lymphatic networks. *Nat. Commun.* **14**, 5878 (2023).
3. Dellinger, M. T., Meadows, S. M., Wynne, K., Cleaver, O. & Brekken, R. A. Vascular endothelial growth factor receptor-2 promotes the development of the lymphatic vasculature. *PloS One* **8**, e74686 (2013).
4. Tvorogov, D. *et al.* Effective suppression of vascular network formation by combination of antibodies blocking VEGFR ligand binding and receptor dimerization. *Cancer Cell* **18**, 630–640 (2010).
5. Deng, Y., Zhang, X. & Simons, M. Molecular controls of lymphatic VEGFR3 signaling. *Arterioscler. Thromb. Vasc. Biol.* **35**, 421–429 (2015).
6. Yao, L.-C. *et al.* Pulmonary lymphangiectasia resulting from vascular endothelial growth factor-C overexpression during a critical period. *Circ. Res.* **114**, 806–822 (2014).
7. Karaman, S. *et al.* Interplay of vascular endothelial growth factor receptors in organ-specific vessel maintenance. *J. Exp. Med.* **219**, e20210565 (2022).
8. Korhonen, E. A. *et al.* Lymphangiogenesis requires Ang2/Tie/PI3K signaling for VEGFR3 cell-surface expression. *J. Clin. Invest.* **132**, e155478 (2022).